# Assessing Urban Sustainability and the Potential to Improve the Quality of Education and Gender Equality in Phnom Penh, Cambodia

Puthearath Chan [1,2,*], Kulakhmetova Gulbaram [3] and Thorsten Schuetze [4,5,*]

1 General Secretariat of the National Council for Sustainable Development, Phnom Penh 12301, Cambodia
2 R&D Department, Advanced Sustainability Institute (ASI), Phnom Penh 12203, Cambodia
3 Department of Recreational Geography and Tourism, Al-Farabi Kazakh National University, Almaty 317-117, Kazakhstan; kulakhmetova.g@kaznu.kz
4 Department of Architecture, Sungkyunkwan University, Suwon 440-746, Republic of Korea
5 Department of Global Smart City, Sungkyunkwan University, Suwon 440-746, Republic of Korea
* Correspondence: pchan1@paragoniu.edu.kh or pchan@ppp-asi.org (P.C.); t.schuetze@skku.edu (T.S.)

**Abstract:** This research assessed the urban sustainability of all 14 districts of the Cambodian capital Phnom Penh to identify weaknesses and improvement potentials to achieve the national development goals; the New Urban Agenda (NUA); and the Sustainable Development Goals (SDGs) 11 (sustainable cities and communities), 4 (quality education), and 5 (gender equality). The indicators' selection was based on available data. The analysis of the indicators and their weights was based on the analytic hierarchy process (AHP). Indicator weights were used to improve assessment accuracy and identify each district's unique characteristics and specific strengths and weaknesses. The normal distribution model was used to standardize the variables before comparison. Among the quality education indicators, the access to education and vocational training obtained the highest weight of 0.38, followed by education staff with 0.33 and facilities with 0.29. Among gender-equality-related indicators, the indicators related to professions obtained the highest weight with 0.34, followed by schools with 0.33 and decision-making with 0.32. The most sustainable district was Boeng Keng Kong, with a consolidated result of 22.81 for quality education and gender equality assessment based on indicator weights, followed by the districts Doun Penh with 20.51, Prampir Makara with 19.95, and Chamkarmon with 19.75. This research identified district-specific strengths and weaknesses, whereas the weak points unveil the improvement potential of specific districts.

**Keywords:** New Urban Agenda; urban sustainability assessment; Sustainable Development Goals; SDG 4 quality education; SDG 5 gender equality; SDG 11 sustainable cities and communities; Phnom Penh; Cambodia

## 1. Introduction

Since 2012, more than 50 percent of the world's population has been living in urban areas [1]. According to the projections of the United Nations, globally, the urban population is increasing while the rural population is decreasing, and by 2050, nearly 70 percent of the world population is expected to live in urban areas [2,3]. The projections showed that with the gradual shift of population from rural to urban areas, 2.5 billion people will possibly be added to urban areas by 2050, with close to 90 percent of this increase taking place in Africa and Asia [2]. This means that most of the rapid urbanization is happening in developing countries. According to the Martin Prosperity Institute (2017), Cambodia and six other countries in Southeast Asia, namely Indonesia, Malaysia, Philippines, Singapore, Thailand, and Vietnam, are facing rapid urbanization. The urban population of this region's 280 million people is projected to grow by another 100 million people by 2030 [4]. Therefore, improving urban quality and sustainability is essential, particularly in African and Southeast Asian countries.

Towards improving urban quality and sustainability, the United Nations committed and set a specific goal for sustainable cities and communities in 2015 under the established 17 Sustainable Development Goals (SDGs) [5]. The goals were adopted to be achieved by 193 nations, including Cambodia, to end poverty and create continuous peace and prosperity for the planet and people [6]. The goal for SDG 11, sustainable cities and communities, aims to make cities and human settlements inclusive, safe, resilient, and sustainable. Notably, its target 11.a aims to support positive economic, social, and environmental links between urban, peri-urban, and rural areas by strengthening national and regional development planning [7]. Furthermore, many studies have shown that SDG 4 and 5, quality education and gender equality, significantly contribute to achieving SDG 11, sustainable cities and communities [8–12]. However, there is a lack of research on the progress of SDG 11 focused on SDG 4 and 5 in Cambodia, particularly at the district level. At the same time, the Cambodian government has significantly increased the national budget to achieve these goals, especially SDG 4, to improve the quality of education [13–16].

Moreover, the New Urban Agenda (NUA) adopted by 167 countries, including Cambodia, in 2016 became the global standard for sustainable development, including planning, management, and living in cities. The NUA serves as an influential tool for sustainable urban development in both developed and developing countries by offering a series of sustainable urban development standards that aim at the provision of basic services for all citizens and ensuring that all citizens have access to equal opportunities and face no discrimination [17,18]. The basic services provision includes access to housing, safe drinking water and sanitation, nutritious food, healthcare and family planning, education, culture, and communication technologies. It is vitally and repeatedly expressed that everyone has the right to benefit from city services while calling on city authorities to address the needs of women, youth, children, people with disabilities, and other vulnerable groups [17].

Furthermore, NUA is significantly addressing the sustainable development goals for education (SDG 4) [19] and gender equality (SDG 5) [20], indicated in target 11.a of SDG 11 to make cities and communities inclusive, sustainable, and more education- and gender-responsive. Many studies have also shown that education is significantly influencing environmental attitudes and behavior [21–26]. Therefore, education is a core component of sustainable or pro-environmental behavior and affects the long-term development of sustainable cities and communities. According to the OECD (2021), improved gender equality in decision-making and the professions related to urban planning contributes to the optimization of settlements and urban infrastructure investments to meet the needs of all people, particularly underrepresented groups, including women and children [27].

Accordingly, Cambodia's policy priorities were centered on the theme of growth, employment, equity, and efficiency, which were included in the government's development strategy, the so-called Rectangular Strategy [28–30]. Since 2005, the government has implemented the strategy, successfully contributing to fast positive socio-economic development [31], resulting in Cambodia's title as Asia's so-called New Tiger Economy in 2016 [32,33]. Moreover, the government defined human resources development as a top priority since 2014 [34] and has significantly increased the national budget for education sectors [14]. In just five years, the education budget increased 2.6 times, from 278.87 million in 2015 to USD 724.80 million in 2020, while other sectors did not; for example, public work and transportation sectors only increased from USD 71.32 to USD 81.90 million from 2015 to 2020 [35,36]. The investments reflect the importance and contributions of SDGs 4 and 5 to achieving SDG 11, primarily by education and gender equality to target 11.a and the NUA. The Cambodian government has put much effort into improving quality education and straightening gender equality. The national development strategy set human resource development, including gender equality, as a top priority and has since 2015 significantly increased the associated national budget [34,35].

Therefore, this research aimed to assess urban sustainability, focusing on urban education and gender equality in Phnom Penh based on a framework incorporating the national context and priorities; the NUA; and SDGs 11, 4, and 5. The aim was to identify weaknesses

and improvement potentials. Phnom Penh has a population of more than two million and challenges improving urban sustainability [37,38]. Therefore, this research was executed on district (Khmer: khan) levels to identify and measure the strong and weak sustainability points and improvement potential by indicator weights.

This research included the following: (i) A literature analysis on urban sustainability development at the district level associated with SDG 11-target 11.a, NUA, and SDGs 4 and 5. (ii) Verifying the importance of urban education and gender equality indicators (indicator weights/priorities) in the context of urban sustainability assessment and the framework of sustainability indicators. (iii) Exploring the 14 capital districts in education and vocational training accesses; hygiene and clean water facilities; education staff sufficiency; and gender equality in access to education and vocational training, profession, and decision-making. (iv) Demonstrating the strengths and weaknesses of all 14 districts by indicators and the potential for improvement. (v) Introducing a standard establishment method for measuring the strong and weak points of cities to future research in Cambodia or other countries to use a normal distribution model to standardize the variables before comparison.

## 2. Materials and Methods

### 2.1. Assessment Framework

The assessment framework of this research is based on the national context and priorities indicated in Cambodia's RS and National Strategic Development Plan [31,34], the NUA (providing basic services for all citizens and ensuring all citizens have access to equal opportunities and face no discrimination) [17,18], SDG 11 Target 11.a (supporting positive economic, social, and environmental links between urban, peri-urban, and rural areas by strengthening regional and national development planning) [5,7], and related targets of sustainable development goals for education and gender equality (SDGs 4, 5, and 11, Figure 1) [19,20].

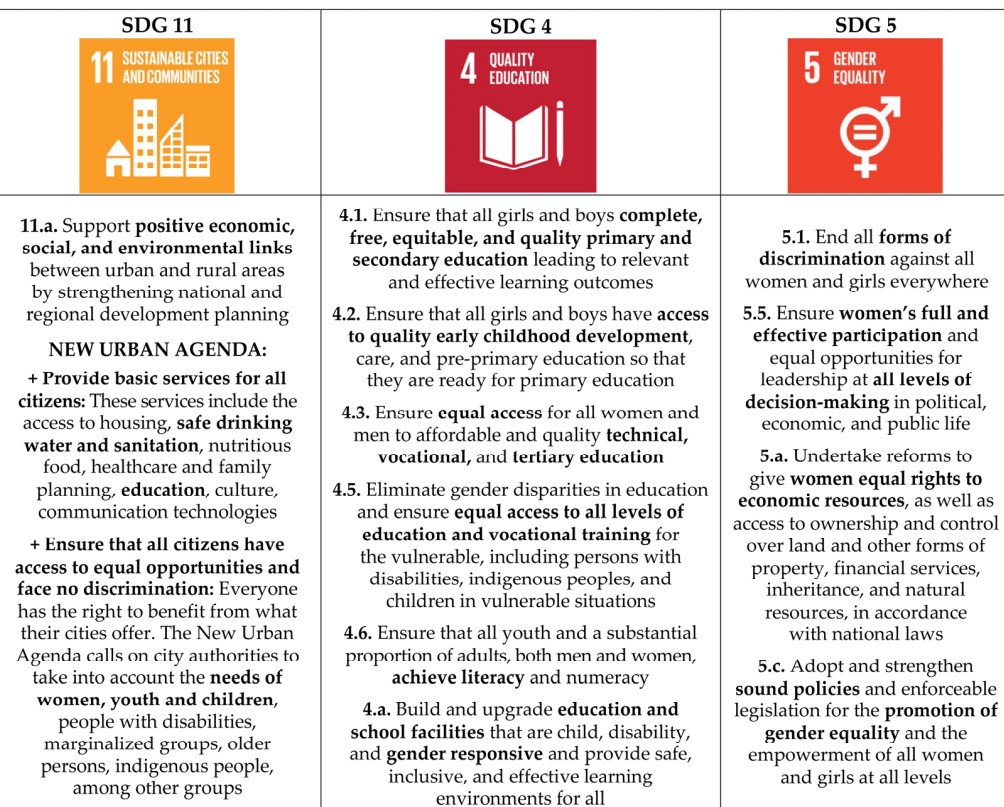

**Figure 1.** SDGs 11, 4, and 5 and associated targets addressed in the assessment framework within this research.

According to Figure 1, the education and gender equality goals associated with the targets of SDG 11 (make cities and human settlements inclusive, safe, resilient, and sustainable) contribute to the urban quality of life [39]. The SDGs are inherently interrelated. Effective, comprehensive action taken towards one goal can support the achievement of other goals. SDGs 4 and 11 are both crucial for long-lasting impact and sustainable development, visible in the indicators applied to SDG 4 and related to gender equality (SDG 5) [40]. SDGs 5 and 11 also significantly support each other. Better representation of women in decision-making and professions related to urban planning could help make cities and settlements more women-sensitive and, in turn, help optimize infrastructure investments to meet the needs of all people [27].

By realizing the requirements of global standards for sustainable urban development and supporting the implementation of sustainable development goals for cities (SDG 11), in particular Target 11.a, the NUA serves as a powerful tool to promote sustainable urban development by providing a series of sustainable urban development standards. The NUA presented a model change based on the knowledge of cities and laid out principles and standards for the planning, development, construction, improvement, and management of urban communities and areas [17,18]. Moreover, the NUA is envisioned as a resource for diverse actors at various levels of government and for NGOs and civil societies, private sectors, and all residing in urban spaces and communities. The NUA highlighted linkages between job creation and sustainable urbanization, employment opportunities, and improved urban quality of life and insisted on the integration of all these sectors in every development of urban policy and strategy [18]. More importantly, the first two main objectives of the NUA aim to provide basic services for all citizens and ensure all citizens have access to equal opportunities and face no discrimination [17], which are quite significant objectives in terms of improving urban education, such as access to general and vocational education and training, sanitation infrastructure in schools, and sufficiency of education staff and strengthening gender equality in both educational access and profession, and in all levels of decision-making.

The assessment indicator framework incorporated the Cambodian context and national priorities (Figure 2), taking the major aspects of providing basic services for all citizens into account, ensuring all citizens have access to equal opportunities and face no discrimination, as indicated in the NUA and target 11.a of SDG 11. Furthermore, the assessment indicators also incorporated the national priorities- and NUA-based SDGs 4 and 5, focusing on urban education and gender quality.

The assessment indicators incorporated the targets of SDGs 4 and 5 within the national priorities and NUA, as shown in Appendix A1. According to the appendix indicators, the assessment indicators addressed SDG 4 targets 4.1, 4.2, 4.3, 4.5, 4.6, and 4.a, while SDG 5 targets 5.1, 5.5, 5.a, and 5.c. In particular, these indicators mostly addressed SDGs 4 and 5 on target 4.a—build and upgrade education and school facilities that are child and gender-responsive and provide safe environments for all; target 4.5—eliminate gender disparities in education and ensure equal access to all levels of education and vocational training for vulnerable groups, including children in vulnerable situations; and target 5.5—ensure women's full and effective participation and equal opportunities for leadership at all levels of decision-making in political, economic, and public life.

### 2.2. Indicator Weighting

To comparatively assess the urban quality of Phnom Penh's 14 districts focusing on urban education and gender equality, this research developed an indicator framework based on (i) the existing context, policy priorities, and available data in Cambodia; (ii) the goals defined in the NUA and SDG 11 target 11.a; and (iii) the education and gender equality related goals SDGs 4 and 5. Eighteen indicators were specified and applied for urban education assessment. Six indicators were assigned to each of three categories: (i) access (access to education, including vocational education and training), (ii) facility (hygiene and clean-water facilities in schools), and (iii) staff (educational staff in schools and education

administrative and coordination offices). Fifteen indicators were specified for urban gender equality assessment. Five indicators were assigned to each of the three categories: (i) in school (gender equality in the access to education and vocational training and related to hygiene facilities in schools), (ii) in professions (gender equality in professions), and (iii) in decision-making (gender equality in decision-making) (Table 1).

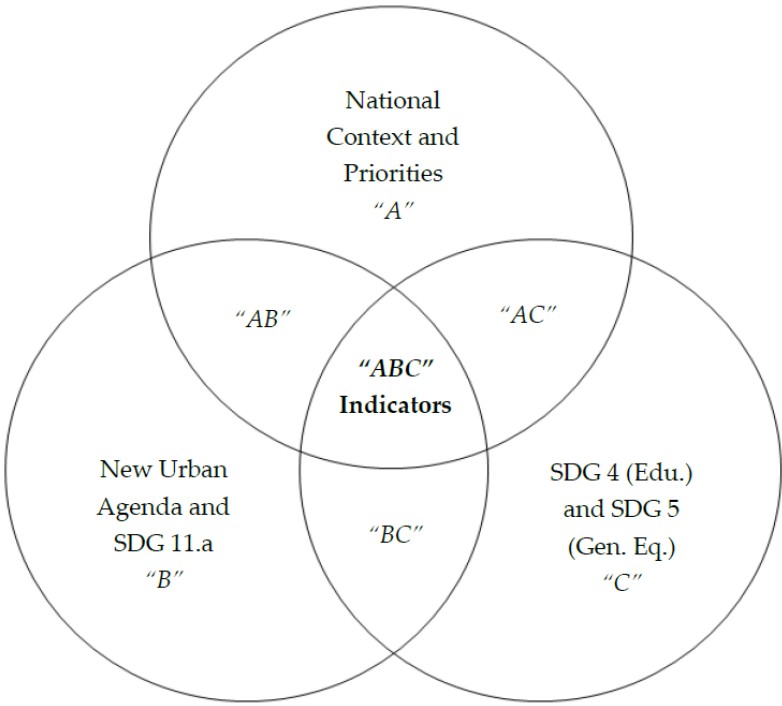

**Figure 2.** Assessment indicator framework.

**Table 1.** All assessment indicators and their assigned acronyms.

| Subject | | Indicator | Category |
|---|---|---|---|
| Urban Education | E01 | Percentage of children studying at kindergarten (aged 3–5) | Access |
| | E02 | Percentage of children studying at primary school (aged 6–11) | |
| | E03 | Percentage of children studying at secondary school (aged 12–14) | |
| | E04 | Percentage of illiterate youth (aged 15–24) | |
| | E05 | Percentage of illiterate adults and middle-aged groups (25–45) | |
| | E06 | Ratio of trained people to people aged 18–35 per 1000 population | |
| | E07 | Ratio of proper toilets installed at primary schools per 100 students | Facility |
| | E08 | Ratio of proper toilets installed in secondary schools per 100 students | |
| | E09 | Ratio of proper toilets installed at high schools per 100 students | |
| | E10 | Percentage of primary schools having clean water to use/drink | |
| | E11 | Percentage of secondary schools having clean water to use/drink | |
| | E12 | Percentage of high schools having clean water to use/drink | |
| | E13 | Ratio of district education administrative office staff per 100,000 population | Staff |
| | E14 | Ratio of district education coordination NGO staff per 100,000 population | |
| | E15 | Ratio of number of primary school students per teacher | |
| | E16 | Ratio of number of secondary school students per teacher | |
| | E17 | Ratio of number of high school students per teacher | |
| | E18 | Percentage of primary and secondary school female teachers | |

**Table 1.** *Cont.*

| Subject | | Indicator | Category |
|---|---|---|---|
| Urban Gender Equality | G01 | Ratio of female students to male students who have studied at university | In school |
| | G02 | Ratio of female students to male students who have studied at high school | |
| | G03 | Ratio of female literates to male literates aged 15–17 years old | |
| | G04 | Ratio of primary schools that have separate toilets for females per 100 students | |
| | G05 | Ratio of secondary schools that have separate toilets for females per 100 students | |
| | G06 | Ratio of the number of vocation-trained women to 100 men aged 18–35 | In professions |
| | G07 | Ratio of female employees to total employees in production-service sectors | |
| | G08 | Percentage of women working at district sectoral technical offices | |
| | G09 | Percentage of female teachers in primary schools | |
| | G10 | Percentage of female teachers in secondary schools | |
| | G11 | Percentage of village chiefs as women | In decision-making |
| | G12 | Percentage of Sankat council members as women | |
| | G13 | Percentage of district council members as women | |
| | G14 | Percentage of district office deputy chiefs as women | |
| | G15 | Percentage of district office chiefs as women | |

Moreover, the analytic hierarchy process (AHP) method has been used to analyze the importance of indicators (indicator weights). The AHP method provides a rational framework for decision-making by quantifying its criteria and alternative options and relating them to the overall goal [41,42]. Thomas L. Saaty founded the AHP in the 1970s [42] and partnered with Ernest Forman to develop Expert Choice in 1983 [43]. AHP is a structured procedure for organizing and analyzing complex decisions, which has been extensively studied and improved [44]. The weights of all indicators are estimated by calculating the principal eigenvector through an AHP matrix. Its consistency is assessed through the consistency index (*CI*) and the consistency ratio (*CR*), as shown in the following Equations (1) and (2):

$$CI = \frac{\lambda max - n}{n - 1} \tag{1}$$

$$CR = \frac{CI}{RI} \tag{2}$$

where *n* is the number of the pairwise-comparison indicators, *λmax* corresponds to the maximum eigenvalue of the comparison matrix, and *RI* is a random consistency index that depends on the number of pairwise-comparison indicators. More importantly, the consistency ratio (*CR*) must be lower than or equal to 0.1 ($CR \leq 0.1$) [42,43].

Reviewed, validated, prioritized, and applied under a framework of the Cambodia Urban Sustainability Assessment (CUSA) project initiated in 2018, the results have been published in an article series [45–48], from which this article is the latest publication. Therefore, the CUSA project analyzed the indicator weights for this article (Table 2) based on its project's indicator prioritization work [48]; the authors' experiences; and in alignment with the theories and research experiences of Soldatou et al. (2022) [44], Dongmin et al. (2020) [49], and other similar studies [50–52]. The CUSA project's prioritization work surveyed and obtained 102 consistent respondents (valid sample size) from offline face-to-face interviews, mainly with government officials, and online surveys via Email, Facebook, and LinkedIn. The surveyed respondents were experienced in working in the fields of sustainable cities and community planning, development, management, and assessment in Cambodia.

**Table 2.** Indicator weights for urban education and gender equality assessment.

| Subject | Category | Weight 1 | Indicator | Weight 2 | Total Weight | Rank |
|---|---|---|---|---|---|---|
| Urban Education | Access | 0.3790 | E01 | 0.1180 | 0.0447 | 12 |
| | | | E02 | 0.2315 | 0.0877 | 1 |
| | | | E03 | 0.1920 | 0.0728 | 3 |
| | | | E04 | 0.1398 | 0.0530 | 9 |
| | | | E05 | 0.1386 | 0.0525 | 10 |
| | | | E06 | 0.1801 | 0.0683 | 6 |
| | Facility | 0.2897 | E07 | 0.1277 | 0.0370 | 16 |
| | | | E08 | 0.1450 | 0.0420 | 14 |
| | | | E09 | 0.2087 | 0.0605 | 8 |
| | | | E10 | 0.2429 | 0.0704 | 4 |
| | | | E11 | 0.1505 | 0.0436 | 13 |
| | | | E12 | 0.1251 | 0.0362 | 17 |
| | Staff | 0.3313 | E13 | 0.1380 | 0.0457 | 11 |
| | | | E14 | 0.1007 | 0.0334 | 18 |
| | | | E15 | 0.2370 | 0.0785 | 2 |
| | | | E16 | 0.2114 | 0.0701 | 5 |
| | | | E17 | 0.1877 | 0.0622 | 7 |
| | | | E18 | 0.1252 | 0.0415 | 15 |
| **Total** | **-** | **1.0000** | **-** | **3.0000** | **1.0000** | **-** |
| Urban Gender Equality | In school | 0.3332 | G01 | 0.2330 | 0.0776 | 2 |
| | | | G02 | 0.2218 | 0.0739 | 3 |
| | | | G03 | 0.2469 | 0.0823 | 1 |
| | | | G04 | 0.1341 | 0.0447 | 15 |
| | | | G05 | 0.1642 | 0.0547 | 13 |
| | In professions | 0.3440 | G06 | 0.2070 | 0.0712 | 7 |
| | | | G07 | 0.2036 | 0.0700 | 9 |
| | | | G08 | 0.1845 | 0.0635 | 11 |
| | | | G09 | 0.1960 | 0.0674 | 10 |
| | | | G10 | 0.2089 | 0.0718 | 5 |
| | In decision-making | 0.3228 | G11 | 0.2197 | 0.0709 | 8 |
| | | | G12 | 0.2272 | 0.0733 | 4 |
| | | | G13 | 0.2218 | 0.0716 | 6 |
| | | | G14 | 0.1752 | 0.0566 | 12 |
| | | | G15 | 0.1561 | 0.0504 | 14 |
| **Total** | **-** | **1.0000** | **-** | **3.0000** | **1.0000** | **-** |

*2.3. Comparative Standard*

Standardizing comparative assessments and using indicator weights are the main methods applied in this research to improve the accuracy of the comparison of Phnom Penh's 14 districts.

Statistical standard variables and standard scores, Z-values, Z-scores, and normal scores, defined the number of standard deviations with the value of a raw score (for

example, the indicator values or data observed) being higher or lower than the mean value of what was observed or measured [46]. The above-average raw scores or raw values resulted in positive standard scores, while the below-average scores or values resulted in negative standard scores, as shown in Figure 3:

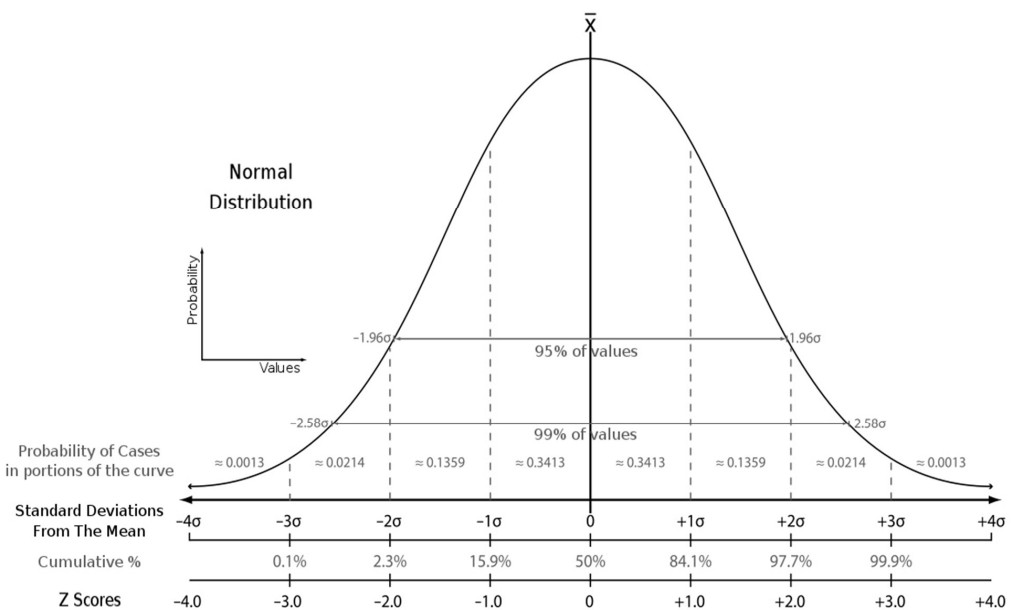

**Figure 3.** Grading methods in a normal distribution. Source: Heds1 and Chan 2020 [46], adapted from Ward and Murray and Claude et al. (2016) [53,54].

The *Z*-values were calculated by subtracting the mean population from each raw score and then dividing the difference by the deviation from the population. The process of converting raw scores into standardized scores is termed normalization; however, normalization can refer to a wide range of ratios. Calculating the *Z*-values required the specification of a mean value (average) and the standard deviation of the full population (variables) with which the data points were associated. If only a sample of population observations is available, then approximation with the sample mean and standard deviation give the *T*-statistics. According to Claude et al. (2016) [54] and Kreyszig (1979) [55], the identified population mean and standard deviation raw score X was converted into a standard score by the following formula, with $\mu$ standing for the mean of the population, and $\sigma$ representing the standard deviation of the population:

$$Z = \frac{X - \mu}{\sigma}$$

The absolute value of *Z* represents the distance between the raw score X and the population mean in units of standard deviation; *Z* is negative when the raw score is below the average and positive when it is above the average. The standard normal distribution tables were also developed by Claude et al. in 2016 and revised by Claude in 2017 [46,54] to provide so-called probability values (*p*-values). According to the developed tables [54], *Z* values (values in the left column and on the top row) are points on the horizontal scale, whereas probabilities (values in the body of the table) are the regions bounded by the normal curve and horizontal scale.

After obtaining the *Z*-values, the 14 Phnom Penh districts were ranked based on the probability values (*p*-values). Accordingly, the ranking resulted in a list from 1 to 14. The scoring values were made based on ranking, which means the 1st rank received 14 scores, the 2nd rank obtained 13 scores, and the 3rd rank obtained 12 scores; this continues until the 13th rank obtained 2 scores, and the 14th (last) rank obtained 1 score. After obtaining the scores of all 14 Phnom Penh districts by each assessment indicator, this research calculated

weighted values (*W*-values) by multiplying the obtained scores by the total weights of the assessment indicators.

### 2.4. Districts and Data Sources

The comparative candidates in this research are the 14 Phnom Penh districts, with a total population of 2,129,371 in 2019 [56]. This capital is located at the intersection of four rivers (four-face river), (i) Sap, Upper Mekong, Lower Mekong, and Bassac. Figure 4 illustrates the capital map with district locations and the population of each district:

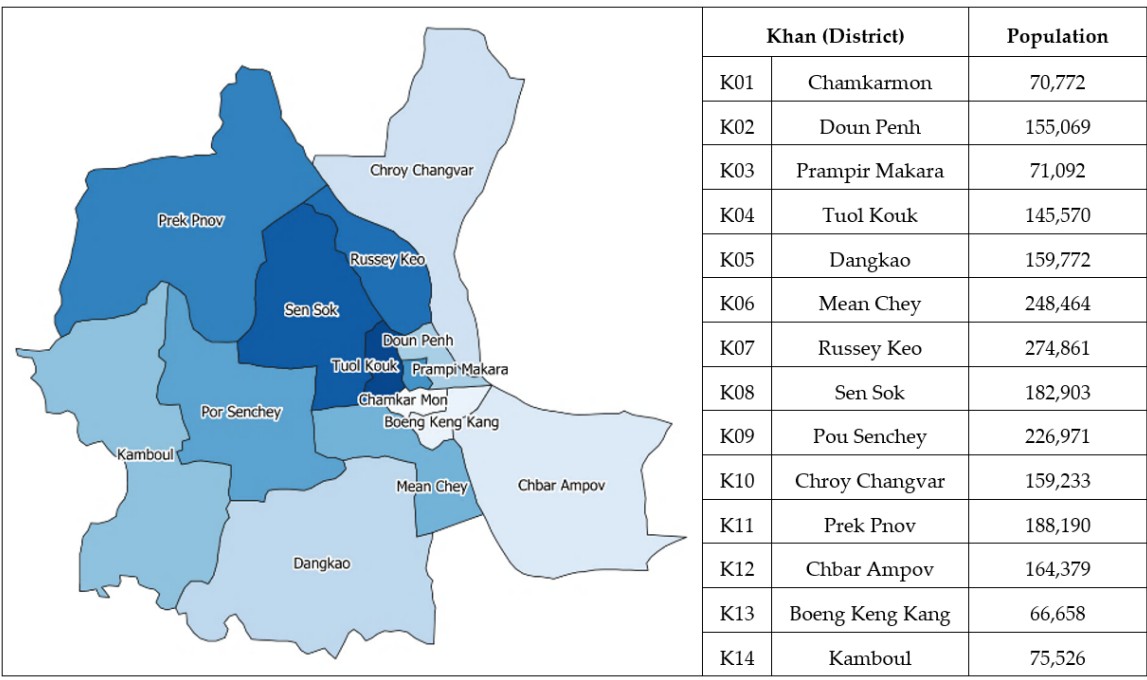

| | Khan (District) | Population |
|---|---|---|
| K01 | Chamkarmon | 70,772 |
| K02 | Doun Penh | 155,069 |
| K03 | Prampir Makara | 71,092 |
| K04 | Tuol Kouk | 145,570 |
| K05 | Dangkao | 159,772 |
| K06 | Mean Chey | 248,464 |
| K07 | Russey Keo | 274,861 |
| K08 | Sen Sok | 182,903 |
| K09 | Pou Senchey | 226,971 |
| K10 | Chroy Changvar | 159,233 |
| K11 | Prek Pnov | 188,190 |
| K12 | Chbar Ampov | 164,379 |
| K13 | Boeng Keng Kang | 66,658 |
| K14 | Kamboul | 75,526 |

**Figure 4.** Populations of the 14 Phnom Penh districts and their location on the capital map. Source: Adapted from Sovan Dara and Socio-Economic Data 2019 [57].

The data of all 14 Phnom Penh districts by each indicator were sourced from the Phnom Penh Capital Socio-Economic Data in 2019 (the government's sources), which was prepared and published by the Phnom Penh Capital Department of Planning under the Ministry of Planning of Cambodia [57]. The data used for all urban education and gender equality assessment indicators were verified, translated from the Cambodian language, and calculated based on the required unit of each assessment indicator.

By using the standard variable methods for the comparative urban assessment, this research is significant in determining the strengths and weaknesses of each Phnom Penh district regarding urban education and gender equality by each assessment indicator and their categories. According to the first [58] and second clean district contests [59], many Phnom Penh districts have been awarded the so-called Clean City Romduol II and III certification. Furthermore, each district has unique characteristics and specific strengths and weaknesses [60]. Identifying, specifying, and quantifying the strengths and weaknesses facilitated the comparison of Phnom Phen districts and the determination of specific improvement potentials. Accordingly, this research developed and applied a standardized method to assess, measure, and compare urban quality focused on urban education and gender equality and in alignment with the research on the development and application of urban sustainability indicators of Han (2019) [61] and Lee (2015) [62]. The final result of this research was the indicator-based quantification, ranking, and specification of each Phnom Penh district's strengths and weaknesses so that an average standard can be used to demonstrate how each city is strong or weak by indicator.

## 3. Results and Discussion

### 3.1. Urban Education Assessment Results

3.1.1. Access to Education, including Vocational Education and Training

The results of the comparative urban education assessments of the 14 Phnom Penh districts regarding access to education, including vocational education and training based on standard values, showed that Prampir Makara (K03) and Tuol Kouk (K04) have higher rates of children studying at kindergarten than the other districts, while Chamkarmon (K01) has a higher rate of children studying at primary school than the other districts. Boeng Keng Kang (K13) has a higher rate of children studying at secondary school than the other districts, while Chroy Changvar (K10) has a higher rate of youth illiteracy than other districts. This district also has a higher illiteracy rate of adults and middle-aged groups than other districts, while Prampir Makara (K03) has higher ratios of populations who have joined and completed vocational education and training than other districts. Table A2 shows the detailed assessment results based on standard values.

Chamkarmon (K01) was found to be the strongest district in urban educational access, followed by Prampir Makara (K03) and Boeng Keng Kong (K13), while Sen Sok (K08) was the weakest district (Figure 5). Chamkarmon (K01) was the strongest because this district has a higher percentage of children studying at both primary and secondary schools and a higher ratio of vocation-trained people per 1000 population. Sen Sok (K08) was the weakest because this district has a lower percentage of children studying at both primary and secondary schools and a higher percentage of illiterate people aged between 25 and 45 years old.

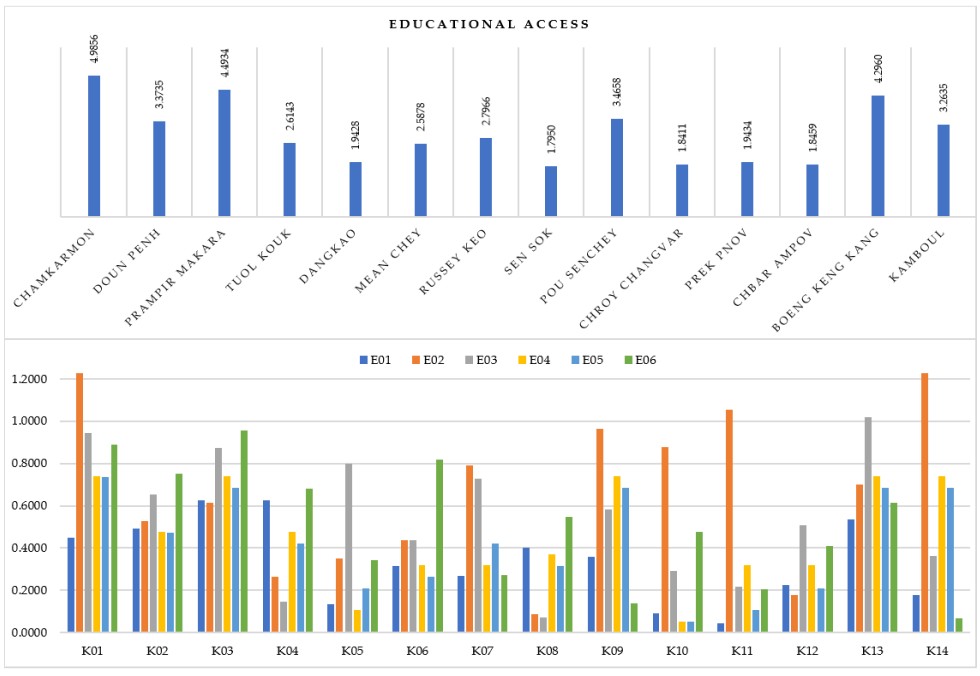

**Figure 5.** Standard value and indicator weight-based assessment results on educational assess.

3.1.2. Hygiene and Clean Water Facilities in Schools

The results of the comparative urban education assessments of 14 Phnom Penh districts on hygiene and clean-water facilities in schools based on standard values showed that Prampir Makara (K03) has a higher ratio of proper toilets installed in primary schools than other districts, while Boeng Keng Kang (K13) has a higher ratio of proper toilets installed in secondary schools than other districts. Prek Pnov (K11) has a higher ratio of proper toilets installed in high schools than other districts, while more than half of all districts, including Chamkarmon (K01), Doun Penh (K02), Prampir Makara (K03), and Boeng Keng Kang (K13), have a higher rate of access to clean water or having clean water to use and drink at all

primary, secondary, and high school levels. Table A3 shows the detailed assessment results based on standard values.

Boeng Keng Kong (K13) was found to be the strongest district in hygiene and clean water facility development in schools, followed by Prek Pnov (K11) and Chroy Changvar (K10), while Chbar Ampov (K12) was the weakest district (Figure 6). The strongest levels of Boeng Keng Kong (K13), Prek Pnov (K11), and Chroy Changvar (K10) were almost similar because these districts have a higher percentage of primary schools and secondary schools having clean water to use and drink, including the percentage of high schools having proper toilets installed per 100 students. Chbar Ampov (K12) was the weakest because this district has a lower percentage of secondary and high schools having clean water to use and drink.

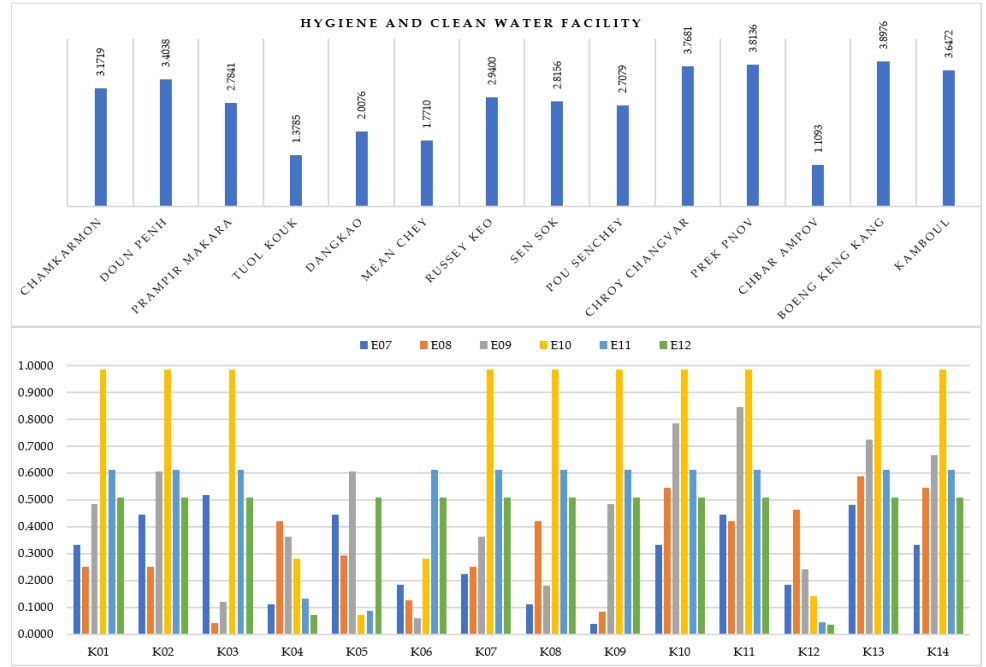

**Figure 6.** Standard value and indicator weight-based assessment results on hygiene–water facility.

### 3.1.3. Educational Staff in Schools and Education Coordination/Administrative Offices

The results of the comparative urban education assessments of 14 Phnom Penh districts on educational staff in schools and education administrative and coordination offices based on standard values showed that Prek Pnov (K11) has a higher ratio of district education administrative office staff per 100,000 population than other districts, while Doun Penh (K02) has a higher ratio of district education coordination NGO staff per 100,000 population than other districts. Kamboul (K14) has a higher ratio of the number of primary school students per teacher than other districts, while Russey Keo (K07) has a higher ratio of the number of secondary school students per teacher than other districts. Pou Senchey (K09) has a higher ratio of the number of high school students per teacher than other districts, while Prampir Makara (K03) has a higher rate of primary and secondary school female teachers than other districts. Table A4 shows the detailed assessment results based on standard values.

Boeng Keng Kong (K13) was found to be the strongest district in the sufficiency of educational staff, followed by Chamkarmon (K01) and Doun Penh (K02), while Pou Senchey (K09) was the weakest district (Figure 7). Boeng Keng Kong (K13) was the strongest because this district has a higher rate of numbers of all primary, secondary, and high school teachers compared to the numbers of students. Pou Senchey (K09) was the weakest because this district has a lower level of all primary, secondary, and high school teachers compared to the number of students.

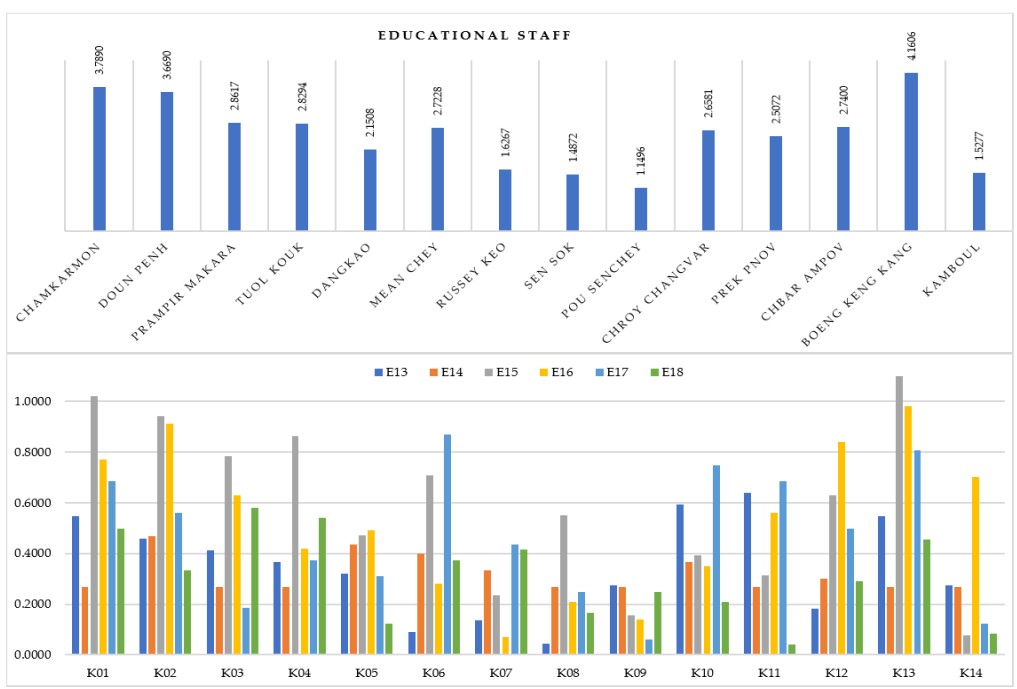

**Figure 7.** Standard value and indicator weight-based assessment results on educational staff.

*3.2. Urban Gender Equality Assessment Results*

3.2.1. Gender Equality in Access to Education and Hygiene Facilities in Schools

The results of the comparative urban gender equality assessments of the 14 Phnom Penh districts on gender equality in access to education and hygiene facilities in schools based on standard values showed that Doun Penh (K02), Russey Keo (K07), and Chbar Ampov (K12) have higher ratios of female students than male students studying at high schools than other districts, while Dangkao (K05) has a higher ratio of female students compared to male students studying at colleges or universities than other districts. Dangkao (K05) has a higher ratio of literate females compared to literate males than other districts, while Prampir Makara (K03) has a higher ratio of separated toilets/developed appropriate toilets for females per 100 students at primary schools than other districts. Boeng Keng Kang (K13) and Kamboul (K14) have higher ratios of separated toilets/developed appropriate toilets for females per 100 students at secondary schools than other districts. Table A5 shows the detailed assessment results based on standard values.

Boeng Keng Kong (K13) was found to be the strongest district in urban gender equality in access to education and hygiene facilities in schools, followed by Sen Sok (K08) and Dangkao (K05), while Pou Senchey (K09) was the weakest (Figure 8). Boeng Keng Kong (K13) was the strongest because this district has a higher ratio of female students compared to male students studying at college or university and a higher ratio of female students than male students studying at high schools; the ratio of female literacy in this district is also higher. Pou Senchey (K09) was the weakest because this district has a lower ratio of primary and secondary schools having separated toilets/developed appropriate toilets for females per 100 students.

3.2.2. Urban Gender Equality in Professions

The results of the comparative urban gender equality assessments of 14 Phnom Penh districts on gender equality in a profession based on standard values showed that Dangkao (K05) has a higher ratio of a number of female populations who have attended vocational education and training than other districts, while Chroy Changvar (K10) has a higher ratio of employees as women to the total employees in production and service sectors than other districts. Chbar Ampov (K12) has a higher rate of women working at the district sectoral technical offices than other districts, while Prampir Makara (K03) has a higher rate

of female teachers in primary and secondary schools than other districts. Table A6 shows the detailed assessment results based on standard values.

Prampir Makara (K03) was found to be the strongest district in gender equality in professions, followed by Mean Chey (K06) and Boeng Keng Kong (K13), while Prek Pnov (K11) was the weakest district (Figure 9). Prampir Makara (K03) was the strongest because this district has a higher percentage of women working at district sectoral technical offices and a higher percentage of female teachers in both primary and secondary schools. Prek Pnov (K11) was the weakest because this district has a lower ratio of female employees to total employees in production and services and a lower percentage of female teachers in both primary and secondary schools.

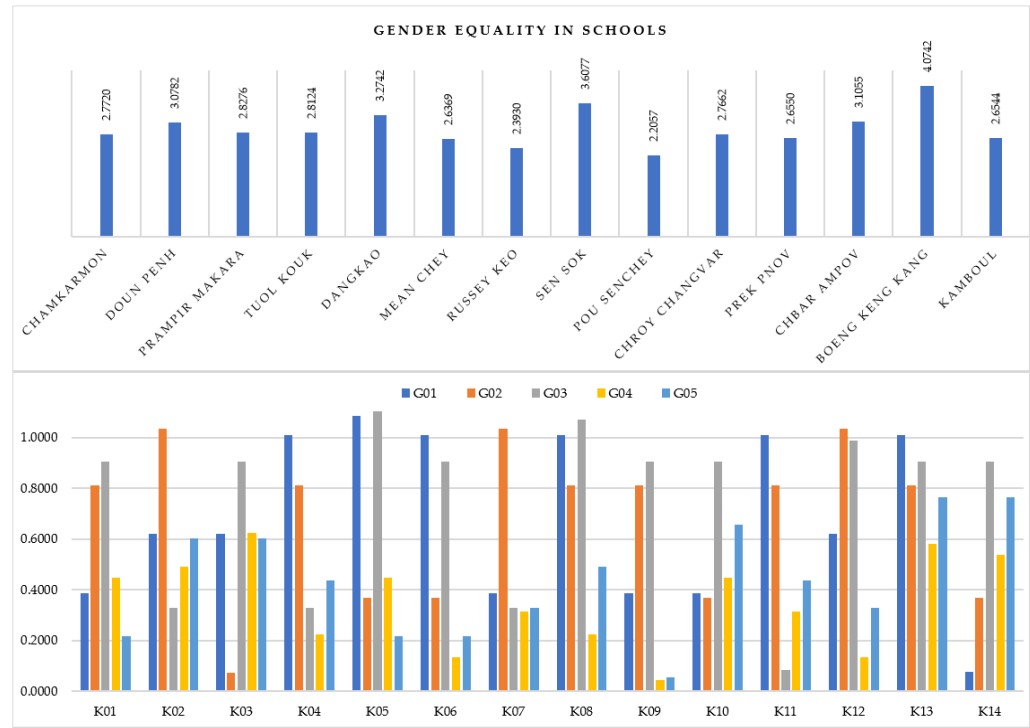

**Figure 8.** Standard value and indicator weight-based assessment results on gender equality in schools.

### 3.2.3. Urban Gender Equality in Decision-Making

The results of the comparative urban gender equality assessments of the 14 Phnom Penh districts on gender equality in decision-making based on standard values showed that Prampir Makara (K03) has a higher rate of village chiefs as women than other districts, while Kamboul (K14) has a higher rate of Sangkat council members as women than other districts. Doun Penh (K02) has a higher rate of both district council members and district office chiefs as women than other districts, while Tuol Kouk (K04) has a higher rate of district office deputy chiefs as women than other districts. Table A7 shows the detailed assessment results based on standard values.

Doun Penh (K02) was found to be the strongest district in urban gender equality in decision-making, followed by Chbar Ampov (K12) and Boeng Keng Kong (K13), while Dangkao (K05) was the weakest district (Figure 10). Doun Penh (K02) was the strongest because this district has a higher percentage of village chiefs as women and Sangkat (commune) council members as women, as well as a higher percentage of district council members as women. Dangkao (K05) was the weakest because this district has a lower percentage of Sangkat (commune) council members as women, as well as a lower percentage of district office chiefs as women.

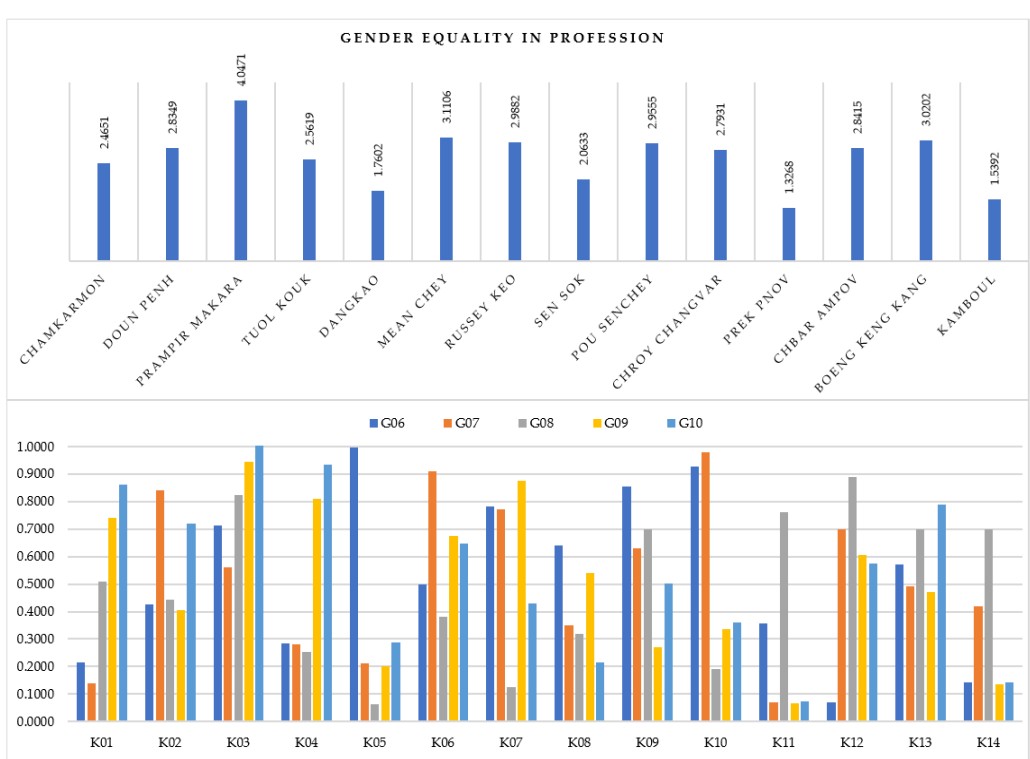

**Figure 9.** Standard value and indicator weight-based assessment results on gender equality in professions.

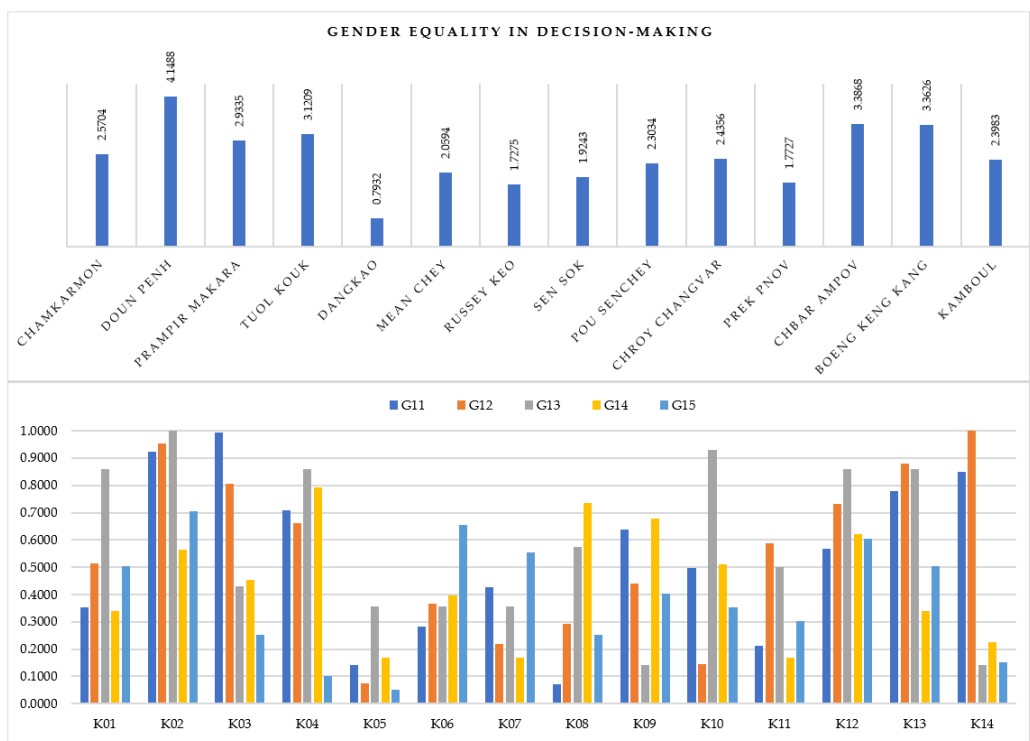

**Figure 10.** Standard value and indicator weight-based assessment results on gender equality in decision-making.

### 3.3. Consolidated Assessment Results

The consolidated results of comparative urban education and gender quality assessment based on indicator weights showed that Boeng Keng Kang (K13) was ranked first for

urban education (12.3541), as shown in Figure 11a. This district furthermore was ranked first for urban gender equality (10.4570), as shown in Figure 11b. Therefore, this district was ranked first overall for consolidated urban education and gender equality (22.8111), as shown in Figure 11c. Chamkarmon (K01) was ranked second for urban education (11.9465) and ranked seventh for urban gender equality (7.8075). In total, this district was ranked fourth for consolidated urban education and gender equality (19.7540). Doun Penh (K02) was ranked third for urban education (10.4463) and ranked second for urban gender equality (10.0619). Overall, this district was ranked second for consolidated urban education and gender equality (20.5082).

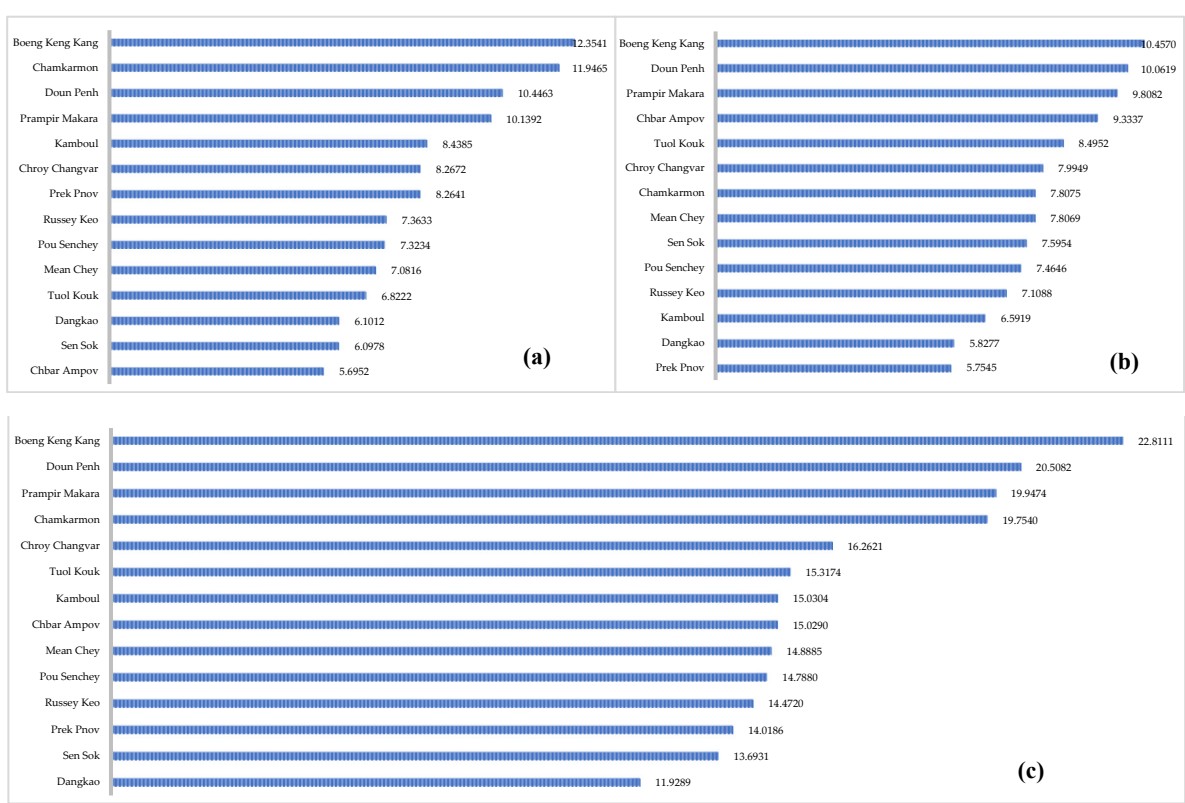

**Figure 11.** Consolidated assessment results on (**a**) urban education, (**b**) urban gender equality, and (**c**) combined urban education and gender equality.

Prampir Makara (K03) was ranked fourth for urban education (10.1392) and third for urban gender equality (9.8082). Overall, this district was ranked third for consolidated urban education and gender equality (19.9474). Kamboul (K14) was ranked fifth for urban education (8.4385) and twelfth for urban gender equality (6.5919). Overall, this capital district was ranked seventh for consolidated urban education and gender equality (15.0304). Chroy Changvar (K10) was ranked sixth for urban education (8.2672) and also sixth for urban gender equality (7.9949). Overall, this district was ranked fifth for consolidated urban education and gender equality (16.2621). Prek Pnov (K11) was ranked seventh for urban education (8.2641) and fourteenth for urban gender equality (5.7545). Overall, this district was ranked twelfth for consolidated urban education and gender equality (14.0186). Russey Keo (K07) was ranked eighth for urban education (7.3633) and eleventh for urban gender equality (7.1088). Overall, this district was ranked eleventh for consolidated urban education and gender equality (14.4720). Pou Senchey (K09) was ranked ninth for urban education (7.3234) and tenth for urban gender equality (7.4646). Overall, this district was ranked tenth for consolidated urban education and gender equality (14.7880). Mean Chey (K06) was ranked tenth for urban education (7.0816) and eighth for urban gender equality (7.8069). Overall, this district was ranked ninth for consolidated urban education and gender equality (14.8555). Tuol Kouk (K04) was ranked eleventh for urban education

(6.8222) and fifth for urban gender equality (8.4952). Overall, this capital district was ranked sixth for consolidated urban education and gender equality (15.3174). Dangkao (K05) was ranked twelfth for urban education (6.1012) and thirteenth for urban gender equality (5.8277). Overall, this capital district was ranked fourteenth for consolidated urban education and gender equality (11.9289). Sen Sok (K08) was ranked thirteenth for urban education (6.0978) and ninth for urban gender equality (7.5954). Overall, this capital district was ranked thirteenth for consolidated urban education and gender equality (13.6931). Finally, Chbar Ampov (K12) was ranked fourteenth for urban education (5.6952) and fourth for urban gender equality (9.3337). Overall, this capital district was ranked eighth for consolidated urban education and gender equality (15.0290).

### 3.4. Strong and Weak Points of Each District

By using the standard variable model for standardizing the comparative assessment of 14 districts of Phnom Penh, the capital of Cambodia by focusing on urban education and gender equality indicators, this research obtained the standard level (orange color), which is significant to find out the strengths and weaknesses 'current level' (blue color) of each district on urban education and gender equality by each assessment indicator. Based on the consolidated assessment results with the urban education and gender equality indicator, the following strengths and weaknesses were revealed for each district (Figures 12–15).

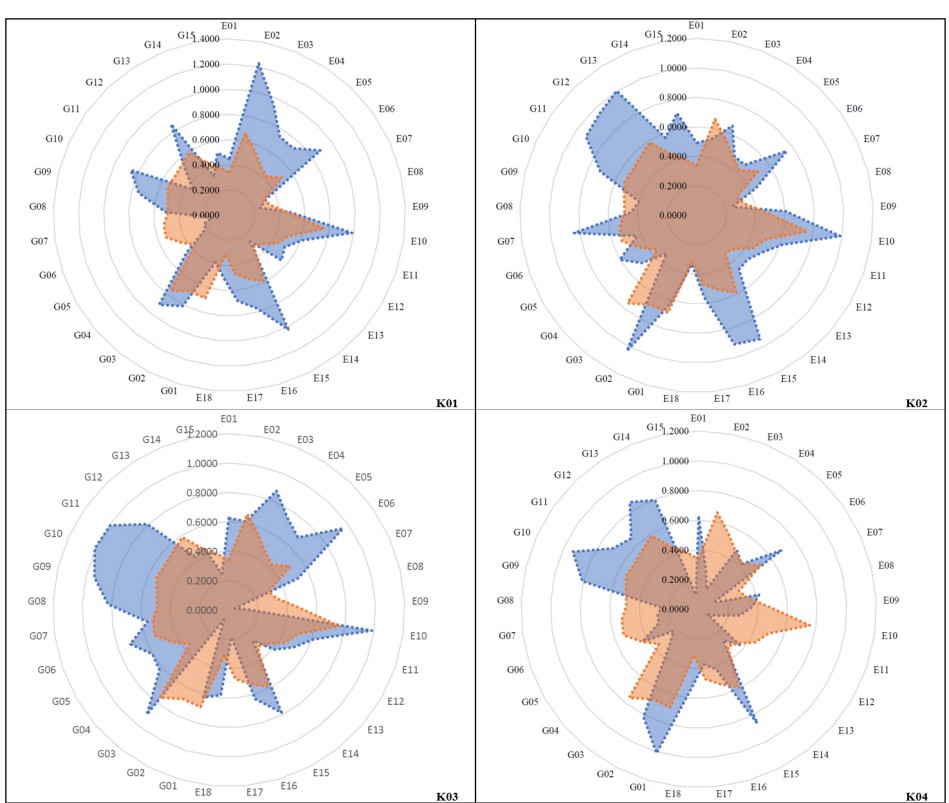

**Figure 12.** Strong and weak points of the districts Chamkarmon (K01), Doun Penh (K02), Prampir Makara (K03), Tuol Kouk (K04) based on the urban education and gender equality indicator.

- Chamkarmon (K01): The consolidated assessment results showed that Chamkarmon was ranked fourth. This district was found to be strong in 24 indicators and weak in 9 indicators, as demonstrated in Figure 12. We observed that the strong points of this district were mainly in educational access (access to education, including vocational education and training) and in the sufficiency of educational staff in schools and education administrative and coordination offices, whereas the weak points of this district were mainly in gender equality in professions.

- Doun Penh (K02): The consolidated assessment results showed that Doun Penh was ranked second. This district was found to be strong in 26 indicators and weak in 7 indicators. We observed that the strong points of this district were mainly in hygiene and clean-water facility development in schools, and in the sufficiency of educational staff in schools and education administrative and coordination offices, as well as in gender equality in decision-making, whereas the weak points of this district were mainly in gender equality in professions.

- Prampir Makara (K03): The consolidated assessment results showed that Prampir Makara was ranked third. This district was found to be strong in 24 indicators and weak in 9 indicators. We observed that the strong points of this district were mainly in educational access (access to education, including vocational education and training) and gender equality in professions, whereas the weak points of this district were mainly in gender equality in access to education and hygiene facilities in schools.

- Tuol Kouk (K04): The consolidated assessment results showed that Tuol Kouk was ranked sixth. This district was found to be strong in 16 indicators and weak in 17 indicators. We observed that the strong points of this district were mainly in the sufficiency of educational staff in schools and education administrative and coordination offices and moderately in gender equality in decision-making, whereas the weak points of this district were mainly in gender equality in access to education and hygiene facilities in schools, and gender equality in professions.

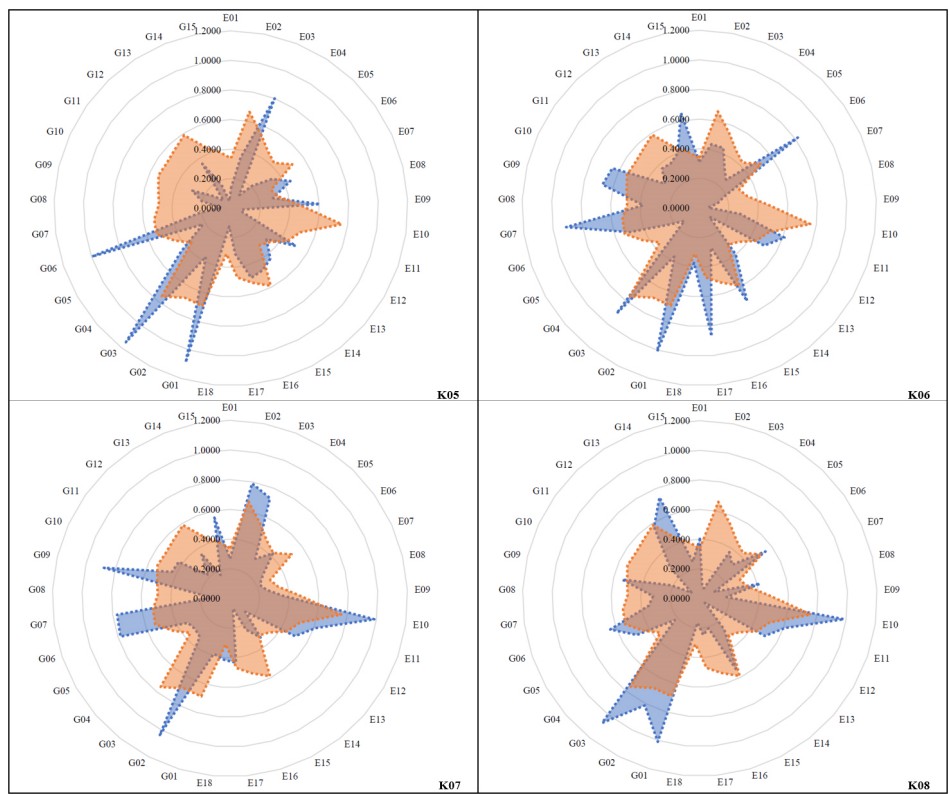

**Figure 13.** Strong and weak points of the districts Dangkao (K05), Mean Chey (K06), Russey Keo (K07), and Sen Sok (K08) based on the urban education and gender equality indicator.

- Dangkao (K05): The consolidated assessment results showed that Dangkao was ranked fourteenth because this district was found to be strong in only 9 indicators but weak in 24 indicators, as demonstrated in Figure 13. We observed that the strong points of this district were mainly in gender equality in access to education and hygiene facilities in schools, whereas the weak points of this district were in almost all indicators of educational access (access to education, including vocational education and training), hygiene and clean-water facility development in schools, the sufficiency of educational

staff in schools and education administrative and coordination offices, and gender equality in professions and decision-making.

- Mean Chey (K06): The consolidated assessment results showed that Mean Chey was ranked ninth. This district was found to be strong in 13 indicators and weak in 20 indicators. We observed that the strong points of this district were mainly in the sufficiency of educational staff in schools and education administrative and coordination offices and in gender equality in professions, whereas the weak points of this district were mainly in educational access (access to education, including vocational education and training), hygiene and clean-water facility development in schools, and gender equality in access to education and hygiene facilities in schools, as well as gender equality in decision-making.

- Russey Keo (K07): The consolidated assessment results showed that Russey Keo was ranked eleventh. This district was found to be strong in 12 indicators and weak in 21 indicators. We observed that the strong points of this district were mainly in gender equality in access to education and hygiene facilities in schools, whereas the weak points of this district were in the sufficiency of educational staff in schools and district education administrative and coordination offices, gender equality in access to education and hygiene facilities in schools, as well as gender equality in decision-making.

- Sen Sok (K08): The consolidated assessment results showed that Sen Sok was ranked thirteenth. This district was found to be strong in 13 indicators and weak in 20 indicators. We observed that the strong points of this district were only gender equality in access to education and hygiene facilities in schools, whereas the weak points of this district were almost all indicators of educational access (access to education, including vocational education and training), hygiene and clean-water facility development in schools, the sufficiency of educational staff in schools and education administrative and coordination offices, gender equality in professions, and gender equality in decision-making.

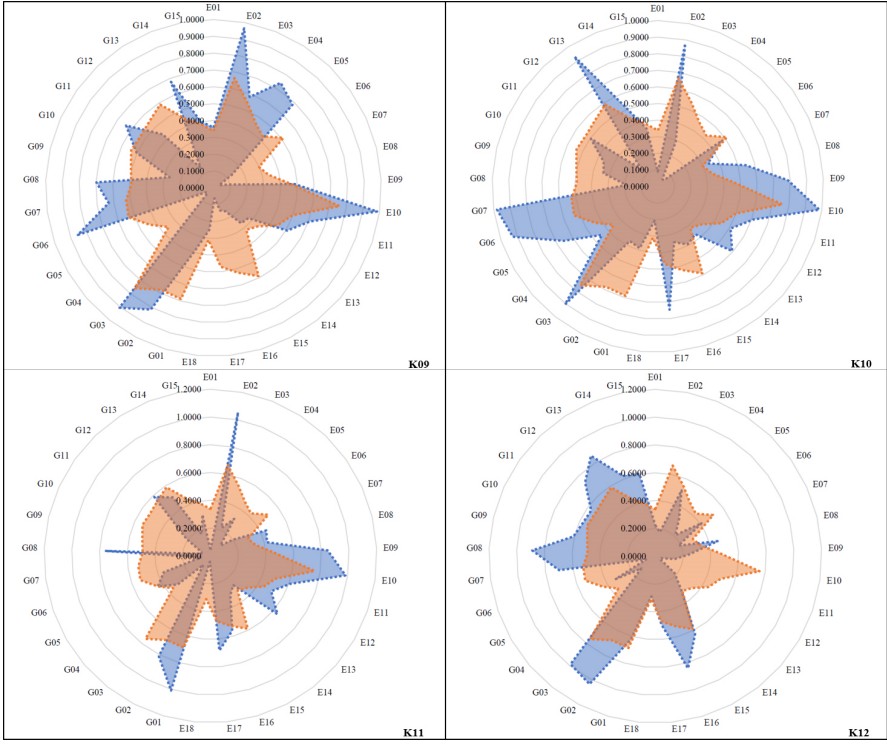

**Figure 14.** Strong and weak points of the districts Pou Senchey (K09), Chroy Changvar (K10), Prek Pnov (K11), and Chbar Ampov (K12) based on the urban education and gender equality indicator.

- Pou Senchey (K09): The consolidated assessment results showed that Pou Senchey was ranked tenth. This district was found to be strong in 17 indicators and weak in 16 indicators, as demonstrated in Figure 14. We observed that the strong points of this district were moderately in educational access (access to education, including vocational education and training) and gender equality in professions, whereas the weak points of this district were mainly in gender equality in decision-making. Its weak points were also moderately in hygiene and clean-water facility development in schools and gender equality in access to education and hygiene facilities in schools.

- Chroy Changvar (K10): The consolidated assessment results showed that Chroy Changvar was ranked fifth. This district was found to be strong in 17 indicators and weak in 16 indicators. We observed that the strong points of this district were mainly in hygiene and clean-water facility development in schools, and moderately in gender equality in access to education and hygiene facilities in schools and gender equality in professions, whereas the weak points of this district were mainly in educational access (access to education, including vocational education and training), and gender equality in professions and decision-making.

- Prek Pnov (K11): The consolidated assessment results showed that Prek Pnov was ranked twelfth. This district was found to be strong in 15 indicators and weak in 18 indicators. We observed that the strong points of this district were only in hygiene and clean-water facility development in schools, whereas the weak points of this district were mainly in educational access (access to education, including vocational education and training), and almost all indicators in gender equality in access to education and hygiene facilities in schools, gender equality in professions, and gender equality in decision-making.

- Chbar Ampov (K12): The consolidated assessment results showed that Chbar Ampov was ranked eighth. This district was found to be strong in 15 indicators and weak in 18 indicators. We observed that the strong points of this district were mainly in gender equality in professions and in decision-making, whereas the weak points of this district were mainly in educational access (access to education, including vocational education and training), and hygiene and clean-water facility development in schools.

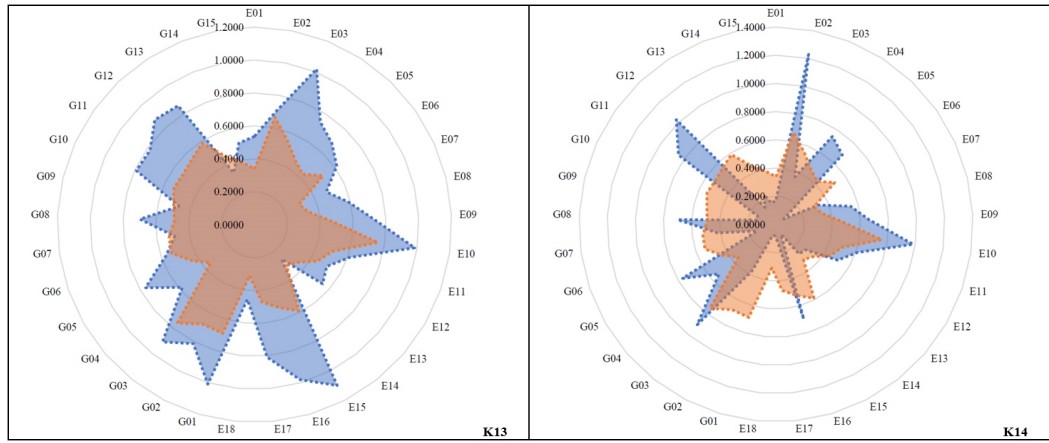

**Figure 15.** Strong and weak points of the districts Boeng Keng Kang (K13) and Kamboul (K14) based on the urban education and gender equality indicator.

- Boeng Keng Kang (K13): The consolidated assessment results showed that Boeng Keng Kang was ranked first. This district was found to be strong in 29 indicators and weak in just 4 indicators, as demonstrated in Figure 15. We observed that the strong points of this district were in almost all indicators of educational access, facilities, and staff, as well as in gender equality in schools, profession, and decision-making, whereas the weak points of this district were only in gender equality in production and service profession, and in decision-making at the district sectoral office levels.

- Kamboul (K14): The consolidated assessment results showed that Kamboul was ranked seventh. This district was found to be strong in 16 indicators and weak in 17 indicators. We observed that the strong points of this district were mainly in hygiene and clean-water facility development in schools and moderately in educational access (access to education, including vocational education and training), whereas the weak points of this district were mainly in educational staff sufficiency, and gender equality in professions.

According to the research findings, the weak points of each district, which accordingly need to be improved, are as follows: Chamkarmon (K01) and Doun Penh (K02) were mainly weak in gender equality in professions, whereas Prampir Makara (K03) was mainly week in gender equality in access to education and hygiene facilities in schools. Tuol Kouk (K04) was particularly weak in gender equality in access to education and hygiene facilities in schools and gender equality in professions, whereas Dangkao (K05) was weak in almost all indicators of educational access (access to education, including vocational education and training), hygiene and clean-water facility development in schools, the sufficiency of educational staff in schools and education administrative and coordination offices, and gender equality in professions and decision-making. Mean Chey (K06) was mainly weak in educational access (access to education, including vocational training), hygiene and clean-water facility development in schools, gender equality in access to education and hygiene facilities in schools, and gender equality in decision-making. In contrast, Russey Keo (K07) was mainly weak in the sufficiency of educational staff in schools and district education administrative and coordination offices, gender equality in access to education and hygiene facilities in schools, and gender equality in decision-making. Sen Sok (K08) was weak in almost all indicators of educational access (access to education, including vocational education and training), hygiene and clean-water facility development in schools, the sufficiency of educational staff in schools and education administrative and coordination offices, gender equality in professions, and gender equality in decision-making. In contrast, Pou Senchey (K09) was mainly weak in gender equality in decision-making and moderately weak in hygiene and clean-water facility development in schools and gender equality in access to education and hygiene facilities in schools. Chroy Changvar (K10) was mainly weak in educational access (access to education, including vocational education and training) and gender equality in professions and decision-making. In contrast, Prek Pnov (K11) was mainly weak in educational access (access to education, including vocational education and training), and almost all indicators in gender equality in access to education and hygiene facilities in schools, gender equality in professions, and gender equality in decision-making. Chbar Ampov (K12) was mainly weak in educational access (access to education, including vocational education and training) and hygiene and clean-water facility development in schools. In contrast, Boeng Keng Kang (K13) was weak at gender equality in the production and service profession and in decision-making at the district sectoral office levels. Finally, Kamboul (K14) was mainly weak in educational staff sufficiency and gender equality.

## 4. Conclusions

Through the comparative urban education and gender equality assessment of the 14 capital districts, this research found that all districts had major strengths and weaknesses around the indicators of education and vocational training access and gender equality in decision-making. The educational access covered the access to childcare and preschool (kindergarten) and primary education, including vocational education and training and the literacy of youth, adults, and middle-aged groups. The gender equality in decision-making on village and district levels covered (i) gender equality at the village management level, (ii) gender equality at the commune management level, (iii) gender equality at the district management levels, and (iv) gender equality at the district office management levels. The following districts showed major strength in gender equality in decision-making: Chamkarmon (K01) was mainly strong in educational access, while Doun Penh (K02) was

mainly strong in gender equality in decision-making. Prampir Makara (K03) was also mainly strong in educational access, while Boeng Keng Kang (K13) was mainly strong in both educational access and gender equality in decision-making. These aspects resulted in these districts obtaining a good rank (top four districts) from the comparative assessment of all capital districts.

This research revealed that most districts still have weaknesses in the indicators of access to quality education and gender equality in decision-making. Therefore, to achieve the national priorities; the global sustainable city and community goals; and the New Urban Agenda, notably Target 11.a, all districts should (i) improve access to both childcare and preschool (kindergarten) and primary education, including the vocational education and training and literacies of youth, adults, and middle-aged groups, and (ii) strengthen gender equality in decision-making from the village level to commune (Sangkat) and district levels, such as gender equality at the village management level (village chiefs), commune management level (council members), district management level (council members), and district sectoral offices' management levels (district office chiefs/deputy chiefs). As gender equality in decision-making in this research is limited to the management levels of public administration, the presented results on urban gender equality in decision-making do not reflect the management levels of non-governmental organizations, the government's development partners, and private companies in each district. Therefore, future research considering these aspects could contribute to an understanding of gender equality in decision-making across those organizations.

**Author Contributions:** Conceptualization, methodology, investigation, and formal analysis, P.C.; validation, P.C., T.S. and K.G.; writing—original draft preparation, P.C.; writing—review and editing, P.C., T.S. and K.G. All authors have read and agreed to the published version of the manuscript.

**Funding:** This research was funded by the CHEY Institute for Advanced Studies through the International Scholar Exchange Fellowship (ISEF) program for the academic year 2021–2022.

**Institutional Review Board Statement:** Not applicable.

**Informed Consent Statement:** Not applicable.

**Data Availability Statement:** Data will be made available upon request.

**Acknowledgments:** The authors acknowledge the support of the CHEY Institute for Advanced Studies. The research results were presented at the Seminar for the 2021–2022 International Scholar Exchange Fellowship hosted by the CHEY Institute for Advanced Studies in Seoul, Korea.

**Conflicts of Interest:** The authors declare no conflict of interest.

## Appendix A  Indicators for Urban Education and Gender Equality Assessment

**Table A1.** Indicators and their relevant sustainable development goals (SDGs) 4 and 5 targets.

| No. | Indicator | Explanation |
|---|---|---|
| I | Access to education, including vocational education and training | |
| 1 | Percentage of children studying at preschool/ kindergartens (aged 3–5) | This indicator assesses early childcare and development through kindergartens; it addresses Target 4.2, and in comparative district assessments, the high percentage of children who have enrolled or studied in this program is much better. |
| 2 | Percentage of children studying at primary schools (aged 6–11) | This indicator assesses the complete and free primary education through the percentage of children that have enrolled or studied in this program, whereas it is currently free for public schools; this indicator addressed Target 4.1. |
| 3 | Percentage of children studying at secondary schools (aged 12–14) | This indicator assesses the complete and free secondary education through the percentage of children who have enrolled or studied in this program, which also addresses Target 4.1, as this program is currently free for public schools. |

**Table A1.** *Cont.*

| No. | Indicator | Explanation |
|---|---|---|
| 4 | Percentage of illiterate youth (aged 15–24) | This indicator assesses the youths' achievements in literacy through the percentage of illiterate youths aged 15 to 24 years old. Therefore, this indicator addresses Target 4.6. |
| 5 | Percentage of illiterate adults and middle-aged groups (aged 25–45) | This indicator assesses the achievement of adults and middle-aged people (aged 25–45) in literacy, which also addresses Target 4.6. In comparative district assessment, a low illiteracy rate is good. |
| 6 | Ratio of trained people to people aged 18–35 per 1,000 population | This indicator assesses vocational education and training through the ratio of trained people to people aged 18–35 per 1,000 population; this indicator addresses Target 4.3. |
| **II** | **Hygiene and clean-water facilities in schools** | |
| 7 | Ratio of proper toilets installed at primary schools per 100 students | This indicator assesses built hygiene facilities in primary schools through the ratio of primary schools having proper toilets installed per 100 students. This indicator addresses Target 4.a. |
| 8 | Ratio of proper toilets installed at secondary schools per 100 students | This indicator assesses built hygiene facilities in secondary schools through the ratio of secondary schools having proper toilets installed per 100 students. This indicator addresses Target 4.a. |
| 9 | Ratio of proper toilets installed at high schools per 100 students | This indicator assesses built hygiene facilities in high schools through the ratio of high schools having proper toilets installed per 100 students. This indicator addresses Target 4.a. |
| 10 | Percentage of primary schools having clean water to use/drink | This indicator assesses clean-water facilities in primary schools through the percentage of primary schools having clean water to use/drink. This indicator addresses Target 4.a. |
| 11 | Percentage of secondary schools having clean water to use/drink | This indicator assesses clean-water facilities in secondary schools through the percentage of secondary schools having clean water to use/drink. This indicator addresses Target 4.a. |
| 12 | Percentage of high schools having clean water to use/drink | This indicator assesses clean-water facilities in high schools through the percentage of high schools having clean water to use/drink. This indicator addresses Target 4.a. |
| **III** | **Education staff in education coordination/administrative offices and schools** | |
| 13 | Ratio of district education administrative office staff per 100,000 population | This indicator assesses the sufficiency of district education office staff to ensure equal access to all levels of education through the ratio of district education administrative office staff per 100,000 population. This indicator addresses Target 4.5. |
| 14 | Ratio of district education coordination NGO staff per 100,000 population | This indicator assesses the sufficiency of district education coordination NGO staff to help and support the inclusion of persons with disabilities, indigenous peoples, and children in vulnerable situations. This indicator addresses Target 4.5. |
| 15 | Ratio of number of primary school students per teacher | This indicator assesses the sufficiency of primary school teachers to support complete primary education through the ratio of the number of students per teacher. This indicator addresses Target 4.1. |
| 16 | Ratio of number of secondary school students per teacher | This indicator assesses the sufficiency of secondary school teachers to support complete and free secondary education through the ratio of the number of students per teacher; consequently, this indicator also addresses Target 4.1. |
| 17 | Ratio of number of high school students per teacher | This indicator assesses the sufficiency of high school teachers to support technical and tertiary education through the ratio of the number of high school students per teacher; therefore, this indicator addresses Target 4.3. |
| 18 | Percentage of primary and secondary school female teachers | This indicator assesses gender equality in education and the adaptation of sound policies for promoting gender equality through the rate of female teachers from primary to secondary schools, which addresses Targets 4.5 and 5.c. |
| **IV** | **Gender equality in access to education and hygiene facilities in schools** | |
| 19 | Ratio of female students to male students studying at university | This indicator assesses gender equality in higher education access and women's equal opportunities through the ratio of female to male students who have studied in university, which addresses both Targets 4.5 and 5.5. |

**Table A1.** *Cont.*

| No. | Indicator | Explanation |
|-----|-----------|-------------|
| 20 | Ratio of female students to male students studying at high schools | This indicator assesses gender equality in education and the adaptation of sound policies for promoting gender equality through the ratio of female to male students who have studied in high school, which addresses Targets 4.5 and 5.c. |
| 21 | Ratio of female literates to male literates aged (15 to 17) years old | This indicator assesses literacy achievement by gender (ending traditional forms of discrimination in education against females) through the ratio of female to male literates. Therefore, this indicator addresses both Targets 4.6 and 5.1. |
| 22 | Ratio of primary schools having separate toilets for females per 100 students | This indicator assesses the adaptation of sound gender policies in built hygiene facilities in primary schools through the ratio of schools having separate toilets for females per 100 students. This indicator addresses Targets 4.a. and 5.c. |
| 23 | Ratio of secondary schools having separate toilets for females per 100 students | This indicator assesses the adaptation of sound gender policies in built hygiene facilities in secondary schools through the ratio of schools having separate toilets for females per 100 students. This indicator also addresses Targets 4.a. and 5.c. |
| V | | **Gender equality in profession** |
| 24 | Ratio of the number of vocation-trained women to 100 men aged (18 to 35) years old | This indicator assesses technical work capacity by gender or women's equal opportunities for vocational education and training through the ratio of the number of vocation-trained women to 100 men aged 18–35. Therefore, this indicator addresses both Targets 4.3 and 5.5. |
| 25 | Ratio of female employees to total employees in production and service sectors | This indicator assesses the employment rate by gender in production and service sectors or women's equal job opportunities in production and services through the ratio of female employees to total employees in production and services. This indicator addresses Target 5.5. |
| 26 | Percentage of women working at district technical offices | This indicator assesses women's full and effective participation and equal opportunities for leadership at the level of decision-making in technical office works through the percentage of women working at district technical offices. This indicator addresses Target 5.5. |
| 27 | Percentage of female teachers in primary schools | This indicator assesses gender equalities in teaching careers, including the adaptation of sound policies for promoting gender equalities through the rate of female teachers in primary schools, which addresses Targets 4.5 and 5.c. |
| 28 | Percentage of female teachers in secondary schools | This indicator assesses gender equalities in teaching careers, including the adaptation of sound policies for promoting gender equalities through the rate of female teachers in secondary schools, which addresses Targets 4.5 and 5.c. |
| VI | | **Gender equality in decision-making** |
| 29 | Percentage of village chiefs as women | This indicator assesses women's full and effective participation and equal opportunities for leadership at the level of decision-making in village-level planning and development through the percentage of village chiefs as women. Thus, this indicator addresses Target 5.5. |
| 30 | Percentage of Sankat council members as women | This indicator assesses women's full and effective participation and equal opportunities for leadership at the level of decision-making in commune-level planning and development through the percentage of Sankat (commune) council members as women. Consequently, this indicator also addresses Target 5.5. |
| 31 | Percentage of district council members as women | This indicator assesses women's full and effective participation and equal opportunities for leadership at the level of decision-making in district-level planning and development through the percentage of district council members as women. This indicator addresses Target 5.5. |
| 32 | Percentage of district sectoral office vice-chiefs as women | This indicator assesses women's full and effective participation and equal opportunities for leadership at the level of decision-making in supporting district sectoral office-level planning and development through the percentage of district sectoral office vice-chiefs as women. Therefore, this indicator addresses Target 5.5. |
| 33 | Percentage of district sectoral office chiefs as women | This indicator assesses women's full and effective participation and equal opportunities for leadership at the level of decision-making in managing district sectoral office-level planning and development through the percentage of district sectoral office chiefs as women. Consequently, this indicator addresses Target 5.5. |

## Appendix B Standard Value-Based Urban Education Assessment Results

**Table A2.** Standard value-based assessment results on urban educational access.

| Ind. | District | Z-Value | p-Value | Rank | Ind. | District | Z-Value | p-Value | Rank | Ind. | District | Z-Value | p-Value | Rank |
|---|---|---|---|---|---|---|---|---|---|---|---|---|---|---|
| E01 | K01 | 0.6927 | 0.7558 | 5 | E03 | K01 | 1.3892 | 0.9176 | 2 | E05 | K01 | −0.9748 | 0.1648 | 14 |
| | K02 | 0.7687 | 0.7790 | 4 | | K02 | 0.1668 | 0.5663 | 6 | | K02 | −0.5921 | 0.2769 | 9 |
| | K03 | 1.1317 | 0.8711 | 1 | | K03 | 0.4936 | 0.6892 | 3 | | K03 | −0.8472 | 0.1984 | 10 |
| | K04 | 1.1317 | 0.8711 | 1 | | K04 | −1.0919 | 0.1374 | 13 | | K04 | −0.2095 | 0.4170 | 7 |
| | K05 | −1.0882 | 0.1382 | 12 | | K05 | 0.4089 | 0.6587 | 4 | | K05 | 0.4282 | 0.6657 | 3 |
| | K06 | 0.2791 | 0.6099 | 8 | | K06 | −0.0268 | 0.4893 | 9 | | K06 | 0.3006 | 0.6181 | 5 |
| | K07 | −0.2779 | 0.3905 | 9 | | K07 | 0.2516 | 0.5993 | 5 | | K07 | −0.2095 | 0.4170 | 7 |
| | K08 | 0.6759 | 0.7504 | 6 | | K08 | −2.1085 | 0.0175 | 14 | | K08 | 0.1731 | 0.5687 | 6 |
| | K09 | 0.6590 | 0.7450 | 7 | | K09 | 0.0821 | 0.5327 | 7 | | K09 | −0.8472 | 0.1984 | 10 |
| | K10 | −1.5356 | 0.0623 | 13 | | K10 | −0.6077 | 0.2717 | 11 | | K10 | 2.2137 | 0.9866 | 1 |
| | K11 | −1.6031 | 0.0545 | 14 | | K11 | −0.8377 | 0.2011 | 12 | | K11 | 1.8311 | 0.9665 | 2 |
| | K12 | −0.7253 | 0.2341 | 10 | | K12 | 0.0458 | 0.5183 | 8 | | K12 | 0.4282 | 0.6657 | 3 |
| | K13 | 0.8616 | 0.8055 | 3 | | K13 | 1.9581 | 0.9749 | 1 | | K13 | −0.8472 | 0.1984 | 10 |
| | K14 | −0.9701 | 0.1660 | 11 | | K14 | −0.1236 | 0.4508 | 10 | | K14 | −0.8472 | 0.1984 | 10 |
| E02 | K01 | 1.0452 | 0.8520 | 1 | E04 | K01 | −0.7932 | 0.2138 | 10 | E06 | K01 | 1.3027 | 0.9037 | 2 |
| | K02 | 0.1720 | 0.5683 | 9 | | K02 | −0.5084 | 0.3056 | 8 | | K02 | 0.7939 | 0.7864 | 4 |
| | K03 | 0.2664 | 0.6050 | 8 | | K03 | −0.7932 | 0.2138 | 10 | | K03 | 1.4348 | 0.9243 | 1 |
| | K04 | −1.1734 | 0.1203 | 12 | | K04 | −0.5084 | 0.3056 | 8 | | K04 | 0.6385 | 0.7384 | 5 |
| | K05 | −0.9373 | 0.1743 | 11 | | K05 | 0.6305 | 0.7358 | 2 | | K05 | −0.6487 | 0.2583 | 10 |
| | K06 | −0.4417 | 0.3294 | 10 | | K06 | 0.3457 | 0.6352 | 3 | | K06 | 1.0242 | 0.8471 | 3 |
| | K07 | 0.4788 | 0.6840 | 6 | | K07 | 0.3457 | 0.6352 | 3 | | K07 | −0.6845 | 0.2468 | 11 |
| | K08 | −1.9050 | 0.0284 | 14 | | K08 | 0.0610 | 0.5243 | 7 | | K08 | 0.4047 | 0.6571 | 7 |
| | K09 | 0.8800 | 0.8106 | 4 | | K09 | −0.7932 | 0.2138 | 10 | | K09 | −1.3522 | 0.0882 | 13 |
| | K10 | 0.7384 | 0.7699 | 5 | | K10 | 2.9083 | 0.9982 | 1 | | K10 | −0.1220 | 0.4514 | 8 |
| | K11 | 0.9036 | 0.8169 | 3 | | K11 | 0.3457 | 0.6352 | 3 | | K11 | −1.1647 | 0.1221 | 12 |
| | K12 | −1.4566 | 0.0726 | 13 | | K12 | 0.3457 | 0.6352 | 3 | | K12 | −0.4399 | 0.3300 | 9 |
| | K13 | 0.3844 | 0.6497 | 7 | | K13 | −0.7932 | 0.2138 | 10 | | K13 | 0.4082 | 0.6584 | 6 |
| | K14 | 1.0452 | 0.8520 | 1 | | K14 | −0.7932 | 0.2138 | 10 | | K14 | −1.5950 | 0.0554 | 14 |

**Table A3.** Standard value-based assessment results on hygiene and clean water facilities.

| Ind. | District | Z-Value | p-Value | Rank | Ind. | District | Z-Value | p-Value | Rank | Ind. | District | Z-Value | p-Value | Rank |
|---|---|---|---|---|---|---|---|---|---|---|---|---|---|---|
| E07 | K01 | −0.2015 | 0.4202 | 6 | E09 | K01 | −0.2011 | 0.4203 | 7 | E11 | K01 | 0.4819 | 0.6851 | 1 |
| | K02 | −0.0840 | 0.4665 | 3 | | K02 | 0.0335 | 0.5134 | 5 | | K02 | 0.4819 | 0.6851 | 1 |
| | K03 | 3.3249 | 0.9996 | 1 | | K03 | −0.9049 | 0.1828 | 13 | | K03 | 0.4819 | 0.6851 | 1 |
| | K04 | −0.5541 | 0.2897 | 12 | | K04 | −0.3184 | 0.3751 | 9 | | K04 | −0.9515 | 0.1707 | 12 |
| | K05 | −0.0840 | 0.4665 | 3 | | K05 | 0.0335 | 0.5134 | 5 | | K05 | −1.9730 | 0.0242 | 13 |
| | K06 | −0.4366 | 0.3312 | 10 | | K06 | −1.3741 | 0.0847 | 14 | | K06 | 0.4819 | 0.6851 | 1 |
| | K07 | −0.3191 | 0.3748 | 9 | | K07 | −0.3184 | 0.3751 | 9 | | K07 | 0.4819 | 0.6851 | 1 |
| | K08 | −0.5541 | 0.2897 | 12 | | K08 | −0.5530 | 0.2901 | 12 | | K08 | 0.4819 | 0.6851 | 1 |
| | K09 | −0.6717 | 0.2509 | 14 | | K09 | −0.2011 | 0.4203 | 7 | | K09 | 0.4819 | 0.6851 | 1 |
| | K10 | −0.2015 | 0.4202 | 6 | | K10 | 0.7373 | 0.7695 | 2 | | K10 | 0.4819 | 0.6851 | 1 |
| | K11 | −0.0840 | 0.4665 | 3 | | K11 | 2.9660 | 0.9985 | 1 | | K11 | 0.4819 | 0.6851 | 1 |
| | K12 | −0.4366 | 0.3312 | 10 | | K12 | −0.4357 | 0.3315 | 11 | | K12 | −2.3764 | 0.0087 | 14 |
| | K13 | 0.5038 | 0.6928 | 2 | | K13 | 0.3854 | 0.6500 | 3 | | K13 | 0.4819 | 0.6851 | 1 |
| | K14 | −0.2015 | 0.4202 | 6 | | K14 | 0.1508 | 0.5599 | 4 | | K14 | 0.4819 | 0.6851 | 1 |

**Table A3.** *Cont.*

| Ind. | District | Z-Value | p-Value | Rank | Ind. | District | Z-Value | p-Value | Rank | Ind. | District | Z-Value | p-Value | Rank |
|---|---|---|---|---|---|---|---|---|---|---|---|---|---|---|
| | K01 | −0.6397 | 0.2612 | 9 | | K01 | 0.5493 | 0.7086 | 1 | | K01 | 0.3888 | 0.6513 | 1 |
| | K02 | −0.6397 | 0.2612 | 9 | | K02 | 0.5493 | 0.7086 | 1 | | K02 | 0.3888 | 0.6513 | 1 |
| | K03 | −1.1995 | 0.1152 | 14 | | K03 | 0.5493 | 0.7086 | 1 | | K03 | 0.3888 | 0.6513 | 1 |
| | K04 | −0.0800 | 0.4681 | 5 | | K04 | −0.5964 | 0.2754 | 11 | | K04 | −1.9455 | 0.0259 | 13 |
| | K05 | −0.2665 | 0.3949 | 8 | | K05 | −2.2685 | 0.0116 | 14 | | K05 | 0.3888 | 0.6513 | 1 |
| | K06 | −0.8263 | 0.2043 | 12 | | K06 | −0.5964 | 0.2754 | 11 | | K06 | 0.3888 | 0.6513 | 1 |
| | K07 | −0.6397 | 0.2612 | 9 | | K07 | 0.5493 | 0.7086 | 1 | | K07 | 0.3888 | 0.6513 | 1 |
| E08 | K08 | −0.0800 | 0.4681 | 5 | E10 | K08 | 0.5493 | 0.7086 | 1 | E12 | K08 | 0.3888 | 0.6513 | 1 |
| | K09 | −1.0129 | 0.1556 | 13 | | K09 | 0.5493 | 0.7086 | 1 | | K09 | 0.3888 | 0.6513 | 1 |
| | K10 | 1.5993 | 0.9451 | 2 | | K10 | 0.5493 | 0.7086 | 1 | | K10 | 0.3888 | 0.6513 | 1 |
| | K11 | −0.0800 | 0.4681 | 5 | | K11 | 0.5493 | 0.7086 | 1 | | K11 | 0.3888 | 0.6513 | 1 |
| | K12 | 0.4798 | 0.6843 | 4 | | K12 | −2.0311 | 0.0211 | 13 | | K12 | −2.7205 | 0.0033 | 14 |
| | K13 | 1.7858 | 0.9629 | 1 | | K13 | 0.5493 | 0.7086 | 1 | | K13 | 0.3888 | 0.6513 | 1 |
| | K14 | 1.5993 | 0.9451 | 2 | | K14 | 0.5493 | 0.7086 | 1 | | K14 | 0.3888 | 0.6513 | 1 |

**Table A4.** Standard value-based assessment results on urban educational staff.

| Ind. | District | Z-Value | p-Value | Rank | Ind. | District | Z-Value | p-Value | Rank | Ind. | District | Z-Value | p-Value | Rank |
|---|---|---|---|---|---|---|---|---|---|---|---|---|---|---|
| | K01 | 0.9053 | 0.8173 | 3 | | K01 | −1.2547 | 0.1048 | 13 | | K01 | −0.3073 | 0.3793 | 10 |
| | K02 | 0.8849 | 0.8119 | 5 | | K02 | −0.9261 | 0.1772 | 12 | | K02 | −0.1882 | 0.4254 | 9 |
| | K03 | 0.2329 | 0.5921 | 6 | | K03 | −0.5399 | 0.2946 | 10 | | K03 | 0.8091 | 0.7908 | 3 |
| | K04 | −0.0524 | 0.4791 | 7 | | K04 | −0.6796 | 0.2484 | 11 | | K04 | 0.1393 | 0.5554 | 6 |
| | K05 | −0.2765 | 0.3911 | 8 | | K05 | 0.2899 | 0.6141 | 6 | | K05 | 0.1839 | 0.5730 | 5 |
| | K06 | −1.4176 | 0.0781 | 13 | | K06 | −0.4249 | 0.3355 | 9 | | K06 | −2.1231 | 0.0169 | 14 |
| | K07 | −1.0712 | 0.1420 | 12 | | K07 | 0.7829 | 0.7831 | 3 | | K07 | 0.0500 | 0.5199 | 7 |
| E13 | K08 | −1.5603 | 0.0593 | 14 | E15 | K08 | 0.2488 | 0.5983 | 7 | E17 | K08 | 0.2286 | 0.5904 | 4 |
| | K09 | −0.4192 | 0.3375 | 9 | | K09 | 1.0540 | 0.8541 | 2 | | K09 | 1.7617 | 0.9609 | 1 |
| | K10 | 1.0276 | 0.8479 | 2 | | K10 | 0.5446 | 0.7070 | 5 | | K10 | −0.6942 | 0.2438 | 12 |
| | K11 | 1.7815 | 0.9626 | 1 | | K11 | 0.5610 | 0.7126 | 4 | | K11 | −0.3073 | 0.3793 | 10 |
| | K12 | −0.5211 | 0.3012 | 11 | | K12 | 0.0681 | 0.5271 | 8 | | K12 | −0.0542 | 0.4784 | 8 |
| | K13 | 0.9053 | 0.8173 | 3 | | K13 | −1.7476 | 0.0403 | 14 | | K13 | −1.1259 | 0.1301 | 13 |
| | K14 | −0.4192 | 0.3375 | 9 | | K14 | 2.0235 | 0.9785 | 1 | | K14 | 1.6277 | 0.9482 | 2 |
| | K01 | −0.3454 | 0.3649 | 7 | | K01 | −0.5960 | 0.2756 | 11 | | K01 | 1.0128 | 0.8444 | 3 |
| | K02 | 3.4423 | 0.9997 | 1 | | K02 | −1.2518 | 0.1053 | 13 | | K02 | 0.2028 | 0.5804 | 7 |
| | K03 | −0.3454 | 0.3649 | 7 | | K03 | −0.3173 | 0.3755 | 9 | | K03 | 1.6958 | 0.9550 | 1 |
| | K04 | −0.3454 | 0.3649 | 7 | | K04 | 0.1909 | 0.5757 | 6 | | K04 | 1.1982 | 0.8846 | 2 |
| | K05 | 0.1103 | 0.5439 | 2 | | K05 | −0.2026 | 0.4197 | 7 | | K05 | −0.7681 | 0.2212 | 12 |
| | K06 | −0.0872 | 0.4652 | 3 | | K06 | 0.5188 | 0.6980 | 4 | | K06 | 0.2614 | 0.6031 | 6 |
| | K07 | −0.2370 | 0.4063 | 5 | | K07 | 2.0270 | 0.9787 | 1 | | K07 | 0.5249 | 0.7002 | 5 |
| E14 | K08 | −0.3454 | 0.3649 | 7 | E16 | K08 | 0.9942 | 0.8399 | 3 | E18 | K08 | −0.5729 | 0.2833 | 11 |
| | K09 | −0.3454 | 0.3649 | 7 | | K09 | 1.4204 | 0.9223 | 2 | | K09 | −0.4266 | 0.3348 | 9 |
| | K10 | −0.1557 | 0.4382 | 4 | | K10 | 0.4040 | 0.6569 | 5 | | K10 | −0.4363 | 0.3313 | 10 |
| | K11 | −0.3454 | 0.3649 | 7 | | K11 | −0.2354 | 0.4070 | 8 | | K11 | −2.0025 | 0.0226 | 14 |
| | K12 | −0.3093 | 0.3786 | 6 | | K12 | −0.9731 | 0.1653 | 12 | | K12 | 0.0613 | 0.5245 | 8 |
| | K13 | −0.3454 | 0.3649 | 7 | | K13 | −1.4321 | 0.0761 | 14 | | K13 | 0.5444 | 0.7069 | 4 |
| | K14 | −0.3454 | 0.3649 | 7 | | K14 | −0.5469 | 0.2922 | 10 | | K14 | −1.2951 | 0.0977 | 13 |

# Appendix C  Standard Value-Based Urban Gender Equality Assessment Results

**Table A5.** Standard value-based assessment results on urban gender equality in schools.

| Ind. | District | Z-Value | p-Value | Rank | Ind. | District | Z-Value | p-Value | Rank | Ind. | District | Z-Value | p-Value | Rank |
|---|---|---|---|---|---|---|---|---|---|---|---|---|---|---|
| G01 | K01 | −0.9389 | 0.1739 | 10 | G03 | K01 | 0.0000 | 0.5000 | 4 | G05 | K01 | −0.7917 | 0.2143 | 11 |
| | K02 | −0.0626 | 0.4750 | 7 | | K02 | −0.5701 | 0.2843 | 11 | | K02 | 0.0776 | 0.5309 | 4 |
| | K03 | −0.0626 | 0.4750 | 7 | | K03 | 0.0000 | 0.5000 | 4 | | K03 | 0.0776 | 0.5309 | 4 |
| | K04 | 0.8137 | 0.7921 | 2 | | K04 | −0.5701 | 0.2843 | 11 | | K04 | −0.3571 | 0.3605 | 7 |
| | K05 | 1.6900 | 0.9545 | 1 | | K05 | 2.2804 | 0.9887 | 1 | | K05 | −0.7917 | 0.2143 | 11 |
| | K06 | 0.8137 | 0.7921 | 2 | | K06 | 0.0000 | 0.5000 | 4 | | K06 | −0.7917 | 0.2143 | 11 |
| | K07 | −0.9389 | 0.1739 | 10 | | K07 | −0.5701 | 0.2843 | 11 | | K07 | −0.5744 | 0.2829 | 9 |
| | K08 | 0.8137 | 0.7921 | 2 | | K08 | 1.1402 | 0.8729 | 2 | | K08 | −0.1397 | 0.4444 | 6 |
| | K09 | −0.9389 | 0.1739 | 10 | | K09 | 0.0000 | 0.5000 | 4 | | K09 | −1.0091 | 0.1565 | 14 |
| | K10 | −0.9389 | 0.1739 | 10 | | K10 | 0.0000 | 0.5000 | 4 | | K10 | 1.5990 | 0.9451 | 3 |
| | K11 | 0.8137 | 0.7921 | 2 | | K11 | −2.2804 | 0.0113 | 14 | | K11 | −0.3571 | 0.3605 | 7 |
| | K12 | −0.0626 | 0.4750 | 7 | | K12 | 0.5701 | 0.7157 | 3 | | K12 | −0.5744 | 0.2829 | 9 |
| | K13 | 0.8137 | 0.7921 | 2 | | K13 | 0.0000 | 0.5000 | 4 | | K13 | 1.8163 | 0.9653 | 1 |
| | K14 | −1.8152 | 0.0347 | 14 | | K14 | 0.0000 | 0.5000 | 4 | | K14 | 1.8163 | 0.9653 | 1 |
| G02 | K01 | 0.2401 | 0.5949 | 4 | G04 | K01 | −0.1794 | 0.4288 | 5 | | | | | |
| | K02 | 1.3604 | 0.9132 | 1 | | K02 | −0.1076 | 0.4571 | 4 | | | | | |
| | K03 | −2.0006 | 0.0227 | 14 | | K03 | 3.4079 | 0.9997 | 1 | | | | | |
| | K04 | 0.2401 | 0.5949 | 4 | | K04 | −0.3946 | 0.3466 | 10 | | | | | |
| | K05 | −0.8803 | 0.1894 | 10 | | K05 | −0.1794 | 0.4288 | 5 | | | | | |
| | K06 | −0.8803 | 0.1894 | 10 | | K06 | −0.4663 | 0.3205 | 12 | | | | | |
| | K07 | 1.3604 | 0.9132 | 1 | | K07 | −0.3229 | 0.3734 | 8 | | | | | |
| | K08 | 0.2401 | 0.5949 | 4 | | K08 | −0.3946 | 0.3466 | 10 | | | | | |
| | K09 | 0.2401 | 0.5949 | 4 | | K09 | −0.5381 | 0.2953 | 14 | | | | | |
| | K10 | −0.8803 | 0.1894 | 10 | | K10 | −0.1794 | 0.4288 | 5 | | | | | |
| | K11 | 0.2401 | 0.5949 | 4 | | K11 | −0.3229 | 0.3734 | 8 | | | | | |
| | K12 | 1.3604 | 0.9132 | 1 | | K12 | −0.4663 | 0.3205 | 12 | | | | | |
| | K13 | 0.2401 | 0.5949 | 4 | | K13 | 0.1794 | 0.5712 | 2 | | | | | |
| | K14 | −0.8803 | 0.1894 | 10 | | K14 | −0.0359 | 0.4857 | 3 | | | | | |

**Table A6.** Standard value-based assessment results on urban gender equality in professions.

| Ind. | District | Z-Value | p-Value | Rank | Ind. | District | Z-Value | p-Value | Rank | Ind. | District | Z-Value | p-Value | Rank |
|---|---|---|---|---|---|---|---|---|---|---|---|---|---|---|
| G06 | K01 | −0.2585 | 0.3980 | 12 | G08 | K01 | 0.0977 | 0.5389 | 7 | G10 | K01 | 1.0121 | 0.8443 | 3 |
| | K02 | −0.1360 | 0.4459 | 9 | | K02 | 0.0559 | 0.5223 | 8 | | K02 | 0.5675 | 0.7148 | 5 |
| | K03 | 0.2877 | 0.6132 | 5 | | K03 | 1.0161 | 0.8452 | 2 | | K03 | 1.4484 | 0.9262 | 1 |
| | K04 | −0.2381 | 0.4059 | 11 | | K04 | −0.1006 | 0.4599 | 11 | | K04 | 1.0876 | 0.8616 | 2 |
| | K05 | 2.1254 | 0.9832 | 1 | | K05 | −2.7099 | 0.0034 | 14 | | K05 | −0.5651 | 0.2860 | 11 |
| | K06 | −0.1105 | 0.4560 | 8 | | K06 | −0.0067 | 0.4973 | 9 | | K06 | 0.2655 | 0.6047 | 6 |
| | K07 | 0.3592 | 0.6403 | 4 | | K07 | −1.1861 | 0.1178 | 13 | | K07 | −0.0785 | 0.4687 | 9 |
| | K08 | −0.0135 | 0.4946 | 6 | | K08 | −0.0589 | 0.4765 | 10 | | K08 | −0.9846 | 0.1624 | 12 |
| | K09 | 0.7880 | 0.7846 | 3 | | K09 | 0.4734 | 0.6820 | 4 | | K09 | −0.0449 | 0.4821 | 8 |
| | K10 | 1.0228 | 0.8468 | 2 | | K10 | −0.4973 | 0.3095 | 12 | | K10 | −0.2882 | 0.3866 | 10 |
| | K11 | −0.1768 | 0.4298 | 10 | | K11 | 0.5778 | 0.7183 | 3 | | K11 | −2.0333 | 0.0210 | 14 |
| | K12 | −1.9737 | 0.0242 | 14 | | K12 | 1.3919 | 0.9180 | 1 | | K12 | 0.0306 | 0.5122 | 7 |
| | K13 | −0.0901 | 0.4641 | 7 | | K13 | 0.4734 | 0.6820 | 4 | | K13 | 0.9534 | 0.8298 | 4 |
| | K14 | −1.5858 | 0.0564 | 13 | | K14 | 0.4734 | 0.6820 | 4 | | K14 | −1.3705 | 0.0853 | 13 |

**Table A6.** *Cont.*

| Ind. | District | Z-Value | p-Value | Rank | Ind. | District | Z-Value | p-Value | Rank | Ind. | District | Z-Value | p-Value | Rank |
|------|----------|---------|---------|------|------|----------|---------|---------|------|------|----------|---------|---------|------|
| G07 | K01 | −0.9794 | 0.1637 | 13 | G09 | K01 | 0.8935 | 0.8142 | 4 | | | | | |
| | K02 | 0.8533 | 0.8033 | 3 | | K02 | −0.2680 | 0.3944 | 9 | | | | | |
| | K03 | −0.1309 | 0.4479 | 7 | | K03 | 1.7979 | 0.9639 | 1 | | | | | |
| | K04 | −0.8436 | 0.1994 | 11 | | K04 | 1.1915 | 0.8833 | 3 | | | | | |
| | K05 | −0.9455 | 0.1722 | 12 | | K05 | −0.9258 | 0.1773 | 12 | | | | | |
| | K06 | 0.9891 | 0.8387 | 2 | | K06 | 0.2254 | 0.5892 | 5 | | | | | |
| | K07 | 0.5479 | 0.7081 | 4 | | K07 | 1.2018 | 0.8853 | 2 | | | | | |
| | K08 | −0.4364 | 0.3313 | 10 | | K08 | −0.0007 | 0.4997 | 7 | | | | | |
| | K09 | 0.2085 | 0.5826 | 6 | | K09 | −0.8435 | 0.1995 | 11 | | | | | |
| | K10 | 2.3806 | 0.9914 | 1 | | K10 | −0.5660 | 0.2857 | 10 | | | | | |
| | K11 | −1.4885 | 0.0683 | 14 | | K11 | −1.7274 | 0.0420 | 14 | | | | | |
| | K12 | 0.3782 | 0.6474 | 5 | | K12 | 0.0918 | 0.5366 | 6 | | | | | |
| | K13 | −0.1988 | 0.4212 | 8 | | K13 | −0.0213 | 0.4915 | 8 | | | | | |
| | K14 | −0.3345 | 0.3690 | 9 | | K14 | −1.0491 | 0.1471 | 13 | | | | | |

**Table A7.** Standard value-based assessment results on urban gender equality in decision-making.

| Ind. | District | Z-Value | p-Value | Rank | Ind. | District | Z-Value | p-Value | Rank | Ind. | District | Z-Value | p-Value | Rank |
|------|----------|---------|---------|------|------|----------|---------|---------|------|------|----------|---------|---------|------|
| G11 | K01 | −0.5414 | 0.2941 | 10 | G13 | K01 | 0.6753 | 0.7502 | 3 | G15 | K01 | 0.2218 | 0.5878 | 5 |
| | K02 | 1.7300 | 0.9582 | 2 | | K02 | 1.7773 | 0.9622 | 1 | | K02 | 2.0735 | 0.9809 | 1 |
| | K03 | 2.2074 | 0.9864 | 1 | | K03 | −0.4666 | 0.3204 | 9 | | K03 | −0.8316 | 0.2028 | 10 |
| | K04 | −0.0805 | 0.4679 | 5 | | K04 | 0.6753 | 0.7502 | 3 | | K04 | −1.0090 | 0.1565 | 13 |
| | K05 | −1.0023 | 0.1581 | 13 | | K05 | −0.7322 | 0.2320 | 10 | | K05 | −1.1199 | 0.1314 | 14 |
| | K06 | −0.7307 | 0.2325 | 11 | | K06 | −0.7322 | 0.2320 | 10 | | K06 | 1.7021 | 0.9556 | 2 |
| | K07 | −0.3110 | 0.3779 | 9 | | K07 | −0.7322 | 0.2320 | 10 | | K07 | 0.4546 | 0.6753 | 4 |
| | K08 | −1.0517 | 0.1465 | 14 | | K08 | −0.0285 | 0.4887 | 7 | | K08 | −0.8316 | 0.2028 | 10 |
| | K09 | −0.1958 | 0.4224 | 6 | | K09 | −1.4226 | 0.0774 | 13 | | K09 | 0.0388 | 0.5155 | 7 |
| | K10 | −0.2863 | 0.3873 | 8 | | K10 | 1.4188 | 0.9220 | 2 | | K10 | −0.3936 | 0.3469 | 8 |
| | K11 | −0.8459 | 0.1988 | 12 | | K11 | −0.3604 | 0.3593 | 8 | | K11 | −0.5156 | 0.3031 | 9 |
| | K12 | −0.2287 | 0.4096 | 7 | | K12 | 0.6753 | 0.7502 | 3 | | K12 | 0.8372 | 0.7988 | 3 |
| | K13 | 0.2733 | 0.6077 | 4 | | K13 | 0.6753 | 0.7502 | 3 | | K13 | 0.2218 | 0.5878 | 5 |
| | K14 | 1.0634 | 0.8562 | 3 | | K14 | −1.4226 | 0.0774 | 13 | | K14 | −0.8483 | 0.1981 | 12 |
| G12 | K01 | −0.4069 | 0.3420 | 8 | G14 | K01 | −0.4758 | 0.3171 | 9 | | | | | |
| | K02 | 0.2821 | 0.6111 | 2 | | K02 | −0.0228 | 0.4909 | 5 | | | | | |
| | K03 | 0.2062 | 0.5817 | 4 | | K03 | −0.3400 | 0.3669 | 7 | | | | | |
| | K04 | −0.0488 | 0.4805 | 6 | | K04 | 3.1493 | 0.9992 | 1 | | | | | |
| | K05 | −0.6944 | 0.2437 | 14 | | K05 | −0.6573 | 0.2555 | 12 | | | | | |
| | K06 | −0.4612 | 0.3223 | 10 | | K06 | −0.4454 | 0.3280 | 8 | | | | | |
| | K07 | −0.6456 | 0.2593 | 12 | | K07 | −0.6573 | 0.2555 | 12 | | | | | |
| | K08 | −0.5154 | 0.3031 | 11 | | K08 | 0.6116 | 0.7296 | 2 | | | | | |
| | K09 | −0.4394 | 0.3302 | 9 | | K09 | 0.5063 | 0.6937 | 3 | | | | | |
| | K10 | −0.6727 | 0.2506 | 13 | | K10 | −0.1497 | 0.4405 | 6 | | | | | |
| | K11 | −0.2658 | 0.3952 | 7 | | K11 | −0.6573 | 0.2555 | 12 | | | | | |
| | K12 | 0.1682 | 0.5668 | 5 | | K12 | 0.1891 | 0.5750 | 4 | | | | | |
| | K13 | 0.2496 | 0.5985 | 3 | | K13 | −0.4758 | 0.3171 | 9 | | | | | |
| | K14 | 3.2443 | 0.9994 | 1 | | K14 | −0.5748 | 0.2827 | 11 | | | | | |

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
