# Peer review of "Assessing Urban Sustainability and the Potential to Improve the Quality of Education and Gender Equality in Phnom Penh, Cambodia"

_sustainability, doi:10.3390/su15118828_

Round 1
Reviewer 1 Report
Most of my concern regarding this review concerns grammatical issues.
1.
Of the 44 citations one was from 1991 and one was from 1977, one 1987 and one was 2007. One would think much more information of the same caliber could be found.
Grammatical error - all numbers 10 and under are to be written out using letters.
There are 36 instances where the word "that" was unnecessarily used.
Several of the charts were far too small and crowded together to read and decipher.
Although these may be minor corrections, adding them all together makes the article very difficult to easily read without re-reading many sections over again and again and again.
Author Response
Comments Reviewer 1
Submitted: 21 Feb 2023 – 16:21
Received: 27 Mar 2023 – 14:35
Comment 1:
Most of my concern regarding this review concerns grammatical issues. Of the 44 citations one was from 1991 and one was from 1977, one 1987 and one was 2007. One would think much more information of the same caliber could be found.
Answer:
Thank you very much for your time and efforts in reviewing our manuscript and providing thoughtful feedback. We would like to inform you that the citations from 1977 and 1987 are about the background of AHP. So, we cited the original papers of the AHP Inventor, Saaty T.L. Other old references are not cited alone; as you can see in citations [21-26], the mentioned references from 1991 and 2007 are cited together with other recent references. The ideas to strongly confirm that education/raising awareness were found to be influenced on the environmental behaviors since 1991 and currently it is still significant and valid.
Comment 2:
Grammatical error - all numbers 10 and under are to be written out using letters.
Answer:
Thank you for your comment and advise, which we addressed accordingly.
Comment 3:
There are 36 instances where the word "that" was unnecessarily used.
Answer:
Thank you for your comment. We deleted them.
Comment 4:
Several of the charts were far too small and crowded together to read and decipher.
Answer:
Thank you for your comments. We increased the charts' size and separated Figure 12 into four figures to improve the readability.
Comment 5:
Although these may be minor corrections, adding them all together makes the article very difficult to easily read without re-reading many sections over again and again and again.
Answer:
Thank you for your critical comment. Accordingly, we separated Figure 12 into four to improve readability and accessibility. We also assigned the figures closer to the text, explaining the specific content. We hope our improvement is acceptable and thank you very much for your efforts and critical comments.
Reviewer 2 Report
Dear Editor
This study focuses on the quality of education and gender equality in Phnom Penh. The topic is interesting but some changes are needed to recommend it for publication.
Abstract: Authors need to present key findings of their study, particularly those that are connected with the literature.
Introduction: I could not find the theoretical context in which this study belongs. Authors should review the literature on this topic and present this research's importance and the gap in the literature that they intend to address.
Material and methods: It seems that the authors combined this section with the results. I recommend separating these two sections. We need more information about the AHP method, why you selected it among other choices, and how you weighted the indicator. How many experts participated in the weighting process and how they weighted?
Figure 4 needs to become larger. details on the map are not clear. This point is also true for other figures and you need to change the fonts' size and overall sizes.
As anticipated from the introduction, this study is disconnected from the literature. It needs to make a systematic connection with previous studies and compare its results. This study should mention what it adds to the literature and why it is interesting for scholars from others parts of the world.
Author Response
Comments Reviewer 2
Submitted: 22 Feb 2023 – 06:42
Received: 27 Mar 2023 – 14:35
Comment 1:
This study focuses on the quality of education and gender equality in Phnom Penh. The topic is interesting but some changes are needed to recommend it for publication.
Answer:
Thank you very much for your time and efforts in reviewing our manuscript and suggesting important changes to improve our article’s quality for publication.
Comment 2:
Abstract: Authors need to present key findings of their study, particularly those that are connected with the literature.
Answer:
Thank you for your critical comment. Accordingly, we revised our article and the abstract. We included relevant findings from published research and added 19 references to contemporary journal articles to discuss the existing research gap and the importance of our research. Nearly half of the added literature refers to research addressing Cambodia.
Comments 3
Introduction: I could not find the theoretical context in which this study belongs. Authors should review the literature on this topic and present this research's importance and the gap in the literature that they intend to address.
Answer:
Thank you for your comment. We apologize for not presenting these aspects clearly in the previous version. Accordingly, and as explained above, we added 19 references. Please refer for details to the revised text, lines 46-83.
Comment 4:
Material and methods. It seems that the authors combined this section with the results. I recommend separating these two sections. We need more information about the AHP method, why you selected it among other choices, and how you weighted the indicator. How many experts participated in the weighting process and how they weighted?
Answer:
Thank you for your comments and suggestions. We didn’t discuss your requested method due to the detailed explanation in reference [48], which is part of a referenced article series [45−48]. Our submitted article is an additional and the latest article in the series. Based on your comments and clarification of the materials and methods, we added the summarized relevant information to our article. Please refer to the revised text in lines 198-223. As we consider the AHP method information as a supplement to this article don’t want to separate the Materials and Methods section and hope that you can accept our decision.
Comment 5:
Figure 4 needs to become larger. details on the map are not clear. This point is also true for other figures and you need to change the fonts' size and overall sizes.
Answer:
Thank you for your comments. We increased the size and readability of Figure 4 and other figures, especially by separating Figure 12 into four figures.
Comment 6:
As anticipated from the introduction, this study is disconnected from the literature. It needs to make a systematic connection with previous studies and compare its results. This study should mention what it adds to the literature and why it is interesting for scholars from others parts of the world.
Answer:
Thank you for your comments and suggestions here. Accordingly, we revised our introduction. We added 19 new references of relevant journal articles to clarify the research gap and significance of this research. Please see the revised text in lines 46-83.
Reviewer 3 Report
This research assesses the urban quality of all 14 districts (khan) of the Cambodian capital Phnom Penh, focusing on urban education and gender equality to identify weaknesses and potential for improvement.
The subject matter is relevant even if it does not present particular insights of originality
Methodology is very simple and has no particular innovative insights even if applied correctly
Within the conclusions there is a part of discussion that should be inserted earlier
The bibliography is partly limited to Cambodia and contains too many self-citations of authors compared to the total number of papers
Finally, I suggest that the authors examine the literature from a more international context and avoid the numerous self-citations.
Author Response
Comments Reviewer 3
Submitted: 13 Feb 2023 – 09:04
Received: 27 Mar 2023 – 14:35
Comment 1:
This research assesses the urban quality of all 14 districts (khan) of the Cambodian capital Phnom Penh, focusing on urban education and gender equality to identify weaknesses and potential for improvement.
Answer:
Thank you very much for your time and efforts in reviewing our manuscript and providing important feedback to improve our article.
Comment 2:
The subject matter is relevant even if it does not present particular insights of originality.
Answer:
Thank you for your comment. We revised our introduction to present particular insights into the originality of our research. We added 19 references from recent journal articles to improve the literature research and illustrate the research gap and the significance of our research. Please refer to the revised text in lines 46-83.
Comment 3:
Methodology is very simple and has no particular innovative insights even if applied correctly.
Answer:
Thank you for your comment. As we explained in lines 114-122, the assessment framework incorporated the national context and priorities, the NUA, target 11.a of SDG 11, and related targets of SDGs 4 and 5, while the assessment methods took indicator weights into account to improve assessment accuracy. Furthermore, as explained in lines 282-296, this research used each district's unique characteristics and specific strengths and weaknesses to establish a standard for measuring each district's strong and weak points by using a popular normal distribution model to standardize the variables before comparison. The applied method is innovative, elaborated, and used by many researchers.
Comment 4:
Within the conclusions there is a part of discussion that should be inserted earlier.
Answer:
Many thanks for your suggestion. We have separated and moved the relevant content from the Conclusions section to the end of the Results and Discussion section for an improved connection to the findings and a better link to the Conclusions section.
Comment 5:
The bibliography is partly limited to Cambodia and contains too many self-citations of authors compared to the total number of papers.
Answer:
Thank you for your comment. We have expanded the bibliography, mainly referring to Cambodia. Nearly half of the 19 newly added references of recent journal articles are studies in Cambodia.
Please note that this present article is a series of articles under the Cambodia Urban Sustainability Assessment (CUSA) project. This article is the latest in published articles [45−48] and needs to be cited accordingly. However, based on your concerns, we tried to reduce some citations just supporting the literature review.
Comment 6:
Finally, I suggest that the authors examine the literature from a more international context and avoid the numerous self-citations.
Answer:
Thank you for your suggestions. We revised our introduction by adding a literature examination from an international perspective. We added 19 references from actual journal articles, particularly clarifying the research gap and the significance of our research. Please refer for details to lines 46-83. As explained above, most of our self-citations refer to a series of articles, which form the essential condition of the present article and, accordingly, need to be cited. However, we excluded self-citations, only supporting the literature review.
Reviewer 4 Report
I read this article with great interest and would like to thank the editors for allowing me to participate in this work. This article aims to assess the urban quality of 14 districts of the Cambodian capital by focusing on urban education and gender equality. In my opinion, it is a very relevant and important study, especially in the global context. However, while the topic is interesting, some concerns need to be considered by the authors.
-The paper seems more descriptive and a bit repetitive.
-The content doesn't concisely describe and contextualize with respect to the previous and present theoretical background and empirical research (if applicable) on the topic.
-The introduction of the manuscript could be organized better.
-There should be a justifiable discussion in the literature section of your study to develop the interrelation between these goals in order to highlight the gap of the study. To increase the internal validity of the text, it is highly recommended to consider citing more scientific articles on the topic in the introduction part of your study and the discussion section. Only 5/44 scientific articles are added to this topic all other.
-The results and discussion section lacks discussion. The authors should show how the results relate to your literature review.
- The conclusion needs to restructure; some essential information which supposes to be in the conclusion part, is missing. For example, what are the findings to support the study's hypothesis? How you described the contribution of your study to the existing literature? The implications of this study should be highlighted as well. The author may also need to highlight the study's importance in the abstract.
- Text size is different on page 5, lines 177.178
Final Opinion
As indicated in the review, this reviewer suggests a revision for all the above reasons.
Author Response
Comments Reviewer 4
Submitted: 26 Mar 2023 – 21:45
Received: 27 Mar 2023 – 14:35
Comment 1:
I read this article with great interest and would like to thank the editors for allowing me to participate in this work. This article aims to assess the urban quality of 14 districts of the Cambodian capital by focusing on urban education and gender equality. In my opinion, it is a very relevant and important study, especially in the global context. However, while the topic is interesting, some concerns need to be considered by the authors.
Answer:
Thank you very much for your time and efforts in reviewing our manuscript and providing very positive and thoughtful feedback.
Comment 2:
The paper seems more descriptive and a bit repetitive.
Answer:
Thank you for your comment. We accordingly revised our article and avoided repetitions. We reorganized section 3.4 and used four figures instead of only one to improve readability and clarity. We also shortened the explanation text referring to the figures. Furthermore, we separated the descriptive part from the Conclusions section. We revised and moved the text to the end of the Results and Discussion section to improve the link to the Conclusions section.
Comment 3:
The content doesn't concisely describe and contextualize with respect to the previous and present theoretical background and empirical research (if applicable) on the topic.
Answer:
Thank you for your critical remark. We revised our introduction by describing and contextualizing the theoretical background. Therefore, we added 19 references from recent journal articles from international and national perspectives. This revision also clarifies the research gap and the significance of our research. Please refer to the revised text in lines 46-83.
Comment 4:
The introduction of the manuscript could be organized better.
Answer:
Thank you for your comment. As explained above, we have reorganized and improved our Introduction section accordingly.
Comment 5:
There should be a justifiable discussion in the literature section of your study to develop the interrelation between these goals in order to highlight the gap of the study. To increase the internal validity of the text, it is highly recommended to consider citing more scientific articles on the topic in the introduction part of your study and the discussion section. Only 5/44 scientific articles are added to this topic all other.
Answer:
Thank you for your suggestions. We have discussed the literature review and interrelation. We especially cited more scientific articles in the introduction and discussion of the literature review. Thus, we added 19 new references of contemporary journal articles to support the literature and to clarify the research gap and the significance of this research.
Comment 6:
The results and discussion section lacks discussion. The authors should show how the results relate to your literature review.
Answer:
Thank you for your advice. We previously presented the main discussion in the Conclusion section. Based on your suggestions, we have moved and improved the discussion from the Conclusions section to the end of the Results and Discussion section.
Comment 7:
The conclusion needs to restructure; some essential information which supposes to be in the conclusion part, is missing. For example, what are the findings to support the study's hypothesis? How you described the contribution of your study to the existing literature? The implications of this study should be highlighted as well. The author may also need to highlight the study's importance in the abstract.
Answer:
Thank you for your comment. As explained earlier, we revised the conclusion section. The implications of this study are highlighted in the Conclusion section in lines 665-680 and the Abstract in lines 20-26.
Comment 8:
Text size is different on page 5, lines 177.178. As indicated in the review, this reviewer suggests a revision for all the above reasons.
Answer:
Thank you for your observation, and we have revised it accordingly.
We hope you are satisfied with this revision and thank you very much for the opportunity to improve our manuscript.
Round 2
Reviewer 3 Report
None
Author Response
Dear Reviewer,
Thank you very much for your second review and approval.
Best regards,
The Authors